# Restoration of the molecular clock is tumor suppressive in neuroblastoma

Myrthala Moreno-Smith[1], Giorgio Milazzo[2,12], Ling Tao [1,12], Baharan Fekry[3], Bokai Zhu[4], Mahmoud A. Mohammad [5], Simone Di Giacomo [2], Roshan Borkar[6], Karthik Reddy Kami Reddy[6], Mario Capasso[7,8], Sanjeev A. Vasudevan[9], Pavel Sumazin[1], John Hicks[10], Nagireddy Putluri[6], Giovanni Perini[2], Kristin Eckel-Mahan [3], Thomas P. Burris [11] & Eveline Barbieri [1✉]

MYCN activation is a hallmark of advanced neuroblastoma (NB) and a known master regulator of metabolic reprogramming, favoring NB adaptation to its microenvironment. We found that the expression of the main regulators of the molecular clock loops is profoundly disrupted in MYCN-amplified NB patients, and this disruption independently predicts poor clinical outcome. MYCN induces the expression of clock repressors and downregulates the one of clock activators by directly binding to their promoters. Ultimately, MYCN attenuates the molecular clock by suppressing BMAL1 expression and oscillation, thereby promoting cell survival. Reestablishment of the activity of the clock activator RORα via its genetic overexpression and its stimulation through the agonist SR1078, restores BMAL1 expression and oscillation, effectively blocks MYCN-mediated tumor growth and de novo lipogenesis, and sensitizes NB tumors to conventional chemotherapy. In conclusion, reactivation of RORα could serve as a therapeutic strategy for MYCN-amplified NBs by blocking the dysregulation of molecular clock and cell metabolism mediated by MYCN.

[1] Department of Pediatrics, Section of Hematology-Oncology, Texas Children's Cancer and Hematology Centers, Baylor College of Medicine, Houston, TX, USA. [2] Department of Pharmacy and Biotechnology, University of Bologna, Bologna, Italy. [3] Institute of Molecular Medicine, McGovern Medical School at the University of Texas Health Science Center (UT Health), Houston, TX, USA. [4] Department of Medicine, Division of Endocrinology and Metabolism, Aging Institute of UPMC, University of Pittsburgh School of Medicine, Pittsburgh, USA. [5] Department of Pediatrics, Children's Nutrition Research Center, US Department of Agriculture, Agricultural Research Service, Baylor College of Medicine, Houston, TX, USA. [6] Department of Molecular and Cellular Biology, Baylor College of Medicine, Houston, TX, USA. [7] Dipartimento di Medicina Molecolare e Biotecnologie Mediche, Università degli Studi di Napoli, Naples, Italy. [8] CEINGE Biotecnologie Avanzate, Naples, Italy. [9] Division of Pediatric Surgery, Michael E. DeBakey Department of Surgery, Baylor College of Medicine, Houston, TX, USA. [10] Department of Pathology and Immunology, Baylor College of Medicine, Houston, TX, USA. [11] Center for Clinical Pharmacology, Washington University School of Medicine and St. Louis College of Pharmacy, St. Louis, MO, USA. [12] These authors contributed equally: Giorgio Milazzo, Ling Tao. ✉email: exbarbie@txch.org

Oncogenic expression of the MYC family members (e.g., MYC and MYCN) drives many human cancers;[1] however, strategies aiming at chemically disrupting their functions have proven difficult to develop. The MYCN oncogene is amplified in almost half of all high-risk neuroblastomas (NBs) and is the primary oncogene driving this malignancy[2]. Overexpressed MYCN causes spontaneous, high-penetrance NB in mice, originating from neuroblastic precursor cells[3]. In tumors with activated MYC or MYCN, the high demand for growth and biomass accumulation required for tumor progression is achieved by metabolic reprogramming[4]. MYC drives several metabolic pathways associated with cell growth, including increased glucose[5] and glutamine uptake[6], stimulation of mitochondrial biogenesis[7] and nucleotide synthesis[8], and enhanced de novo lipid synthesis[9]. In particular, MYC directly activates the expression of lipogenic enzymes (e.g., ACACA, FASN, and SCD1), promoting de novo fatty acid (FA) synthesis from citrate[10]. In addition, MYC interacts with the related transcriptional regulator MondoA to control SREBP1-dependent lipid biosynthesis[11]. However, far less is known about MYCN regulation of cell metabolism. MYCN loss of function suppresses FA β-oxidation and leads to lipid droplet accumulation in NB cells[12], suggesting that aggressive NBs require FA as an energy source.

Cellular metabolism and circadian rhythm intimately crosstalk[13], and MYC regulates rhythmic cell metabolism[14]. Moreover, genetic ablation of clock genes enhances glucose and glutamine consumption in vivo[15]. The core molecular clock is composed of the heterodimer CLOCK/BMAL1, which activates transcription of the PER and CRY genes via binding at specific E-box sequences. In turn, PER1/2 and CRY1/2 suppress CLOCK/BMAL1[16]. This transcriptional feedback loop ensures a 24-h rhythm. Retinoic acid receptor-related orphan receptor alpha (RORα) and REV-ERBα (also known as nuclear receptor subfamily 1, group D member 1, NR1D1) are the two major regulators of both expression and oscillation of the clock central component BMAL1 (ARNTL) and compete for the ROR response element (RORE) within the BMAL1 promoter[17,18]. RORα activates, whereas REV-ERBα suppresses BMAL1 transcription. MYC and MYCN have been shown to directly induce REV-ERBα expression, dampening BMAL1 expression and oscillation[14]. In addition, MYC/Myc-interacting zinc-finger protein-1 (MIZ1) complexes are recruited to non-E-box sites in the BMAL1 and CLOCK promoters independently of REV-ERB[19]. However, MYCN regulation of the molecular clock and its effects on tumor phenotype remain largely unknown. The clock activator RORα has been identified as a putative tumor suppressor and is repressed in many cancer types[20–23]. Mechanistically, RORα acts by inducing SEMA3F[20] and p53 signaling[24] and suppressing Wnt/βcatenin signaling[25]. Similarly, BMAL1 is epigenetically inactivated in hematologic malignancies[26], and ectopic expression enhances sensitivity to chemotherapy[27] and reduces clonogenicity of NB cells[14], further supporting its potential tumor-suppressive role.

RORs and REV-ERBs are known as orphan nuclear receptors; however, the subsequent identification of their endogenous ligands[28,29] led to the development of synthetic ligands targeting these receptors. Because the clock components can affect tumorigenesis, pharmacological modulation of the circadian clock is emerging as a promising approach for therapeutic intervention. Recent reports found two REV-ERBs agonists (SR9009 and SR9011) were lethal to cancer cells by targeting de novo lipogenesis and autophagy[30]. These clock-modulating compounds are also active against metabolic diseases, including obesity and diabetes[18,31]. The synthetic ligand SR1078 functions as a RORα/γ agonist[29] and selectively binds to the ligand-binding domain of RORα/γ receptors, inducing conformational changes resulting in

recruitment of co-activators and stimulation of transcription[29]. In this study, we asked whether dysregulation of the MYCN-operated clock had functional significance and whether restoring the positive arm of the clock would interfere with that dysregulation, opposing MYCN-driven oncogenesis. Mechanistically, we have shown that MYCN attenuates the clock in NB by down-regulating clock activators and inducing clock repressors. Restoration of the molecular clock reestablishes cellular lipid metabolism and efficiently blocks NB tumor growth.

## Results

**Clock gene expression is altered in MYCN-amplified patients and predicts poor clinical outcomes.** To establish the potential clinical relevance of MYCN regulation of the clock, we examined how the expression of the main clock component (BMAL1) and its activator (RORα) and repressor (REV-ERBα) related to clinical outcome and MYCN amplification in human NB primary tumors. We used transcriptomic data from three patient cohorts: Kocak (n = 649, GEO: GSE45547, patient cohort 1), Versteeg (n = 88, GEO: GSE16476 88/122, patient cohort 2), and NB Research Consortium (NRC; n = 283, GEO: GSE85047, patient cohort 3) (R2: genomics analysis and visualization platform, http://r2.amc.nl). Kaplan–Meier analysis in cohort 1 revealed that high RORα and BMAL1 (ARNTL) expression is strongly associated with a favorable clinical outcome and stage, while high REV-ERBα (NR1D1) expression correlates with a poor outcome and stage (Fig. 1). These findings were validated across multiple patient cohorts (Supplementary Fig. 1a, patient cohort 2) and mark the positive arm of the clock as a potential tumor suppressor. Notably, the expression of both RORα and BMAL1 is strongly repressed in MYCN-amplified (MNA) NB (cohort 1, Fig. 1; cohort 2, Supplementary Fig. 1a), and this correlates with poor overall (OS) and progression-free survival (PFS) in all patient cohorts (Fig. 1 and Supplementary Fig. 1). This association is retained when the analysis is restricted to stage 3 and 4 patients (Supplementary Fig. 1b, c). Conversely, REV-ERBα expression is significantly higher in MNA NB (cohort 1, Fig. 1; cohort 2, Supplementary Fig. 1a). Notably, we found that RORα expression predicts outcome independently of current prognostic factors, including age at diagnosis, International Neuroblastoma Staging System (INSS) stage, and MYCN status (Supplementary Table 1). Collectively, our data define the low expression of the positive arm of the clock (RORα-BMAL1) as an unfavorable prognostic marker in NB.

**MYCN disrupts the molecular clock by direct binding to clock gene promoters.** Due to the significant association between circadian genes and MYCN expression in patients, we next examined whether MYCN could alter clock gene expression. To test this hypothesis, we conditionally overexpressed MYCN in MYCN3 (Tet-ON) cells and conditionally silenced MYCN in LAN5 MNA cells (LAN5 ShMYCN). Ectopic MYCN expression resulted in significant upregulation of the circadian repressor REV-ERBα, and downregulation of the circadian activators RORα and BMAL1. However, depletion of MYCN completely restored their expression levels (Fig. 2a). We then used a human NB line SK-N-AS that lacks amplification of MYCN and stably expresses inducible wild-type MYCN-ER^TM (estrogen receptor tamoxifen mutant). When stimulated with 4-hydroxytamoxifen (4OHT), MYCN-ER translocates into the nucleus and upregulates MYCN targets[14]. MYCN transcriptional activation led to significant upregulation of REV-ERBα and downregulation of RORα and BMAL1 both at mRNA and protein levels (Fig. 2b). As a control, we demonstrated that both doxycycline (DOX) and 4OHT do not alter clock gene expression (DBP, BMAL1, and PER2) in parental

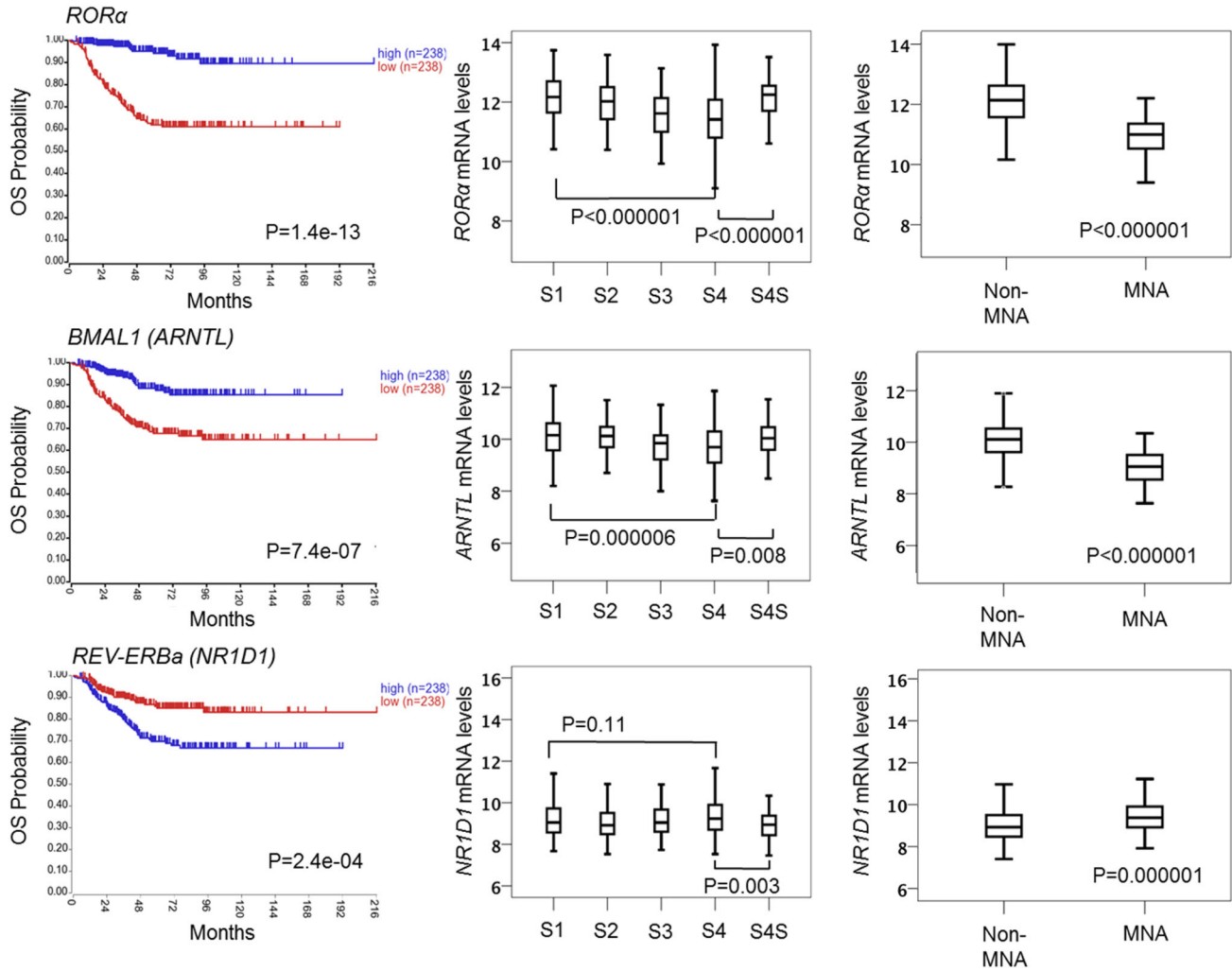

**Fig. 1 Clock gene expression is altered in primary NB tumor samples and correlates with poor clinical outcomes.** Kaplan-Meier analysis of overall survival (OS) in patient cohort 1 (Kocak n = 476, GEO: GSE45547). Graphs depict p-values corrected for multiple testing (Bonferroni correction) of cutoff levels for *RORα*, *BMAL1*, and *REV-ERBα*. Correlations between clock gene expression, INSS stages, and MYCN amplification status are shown. MNA = MYCN-amplified; S1–4S = stage1-4S; 476 of the 649 samples were annotated with survival data. The box plot is defined by two lines at the 25th and 75th percentile. A line is drawn inside the box at the 50th percentile.

LAN5 and SK-N-AS cells (Supplementary Fig. 2). Collectively, these data support a role for MYCN in altering circadian gene expression, and specifically suppressing the positive arm of the clock.

We then asked whether MYCN could also disrupt circadian rhythmicity. We subjected DOX-induced MYCN3 and LAN5 ShMYCN cells to real-time luminescence analysis using a previously established *Per2*-luc adenovirus reporter[32] and found that MYCN induction also impairs the circadian oscillation of *PER2* by both lengthening the circadian period and reducing the circadian amplitude (Supplementary Fig. 3). These results are consistent with previously reported impaired BMAL1 oscillation in MYC overexpressing cell lines[14].

To address the mechanism of MYCN regulation of the clock, we performed MYCN chromatin immunoprecipitation (ChIP)-qPCR analysis in both MNA cells and MYCN Tet-OFF (Tet-21/N) cells, in which MYCN is turned off upon DOX treatment (Fig. 2c and Supplementary Fig. 4). We found significant enrichment of MYCN binding to the promoter regions of *REV-ERBα*, *RORα* (transcript variants 1 and 4), and *BMAL1* in both MNA cell lines. Moreover, turning off MYCN fully abrogates MYCN binding (Fig. 2c). These findings suggest that MYCN

directly attenuates the clock by repressing the clock activators RORα and BMAL1, and inducing the clock repressor REV-ERBα. Because MYCN induces transcriptional repression by indirectly binding to DNA in part through interactions with MIZ1[33], we performed MYCN ChIP-qPCR analysis in LAN5 cells upon genetic depletion of MIZ1 (LAN5 ShMIZ1) to determine whether loss of MIZ1 altered MYCN occupancy at clock activators genes. Effective knockdown of MIZ1 (Supplementary Fig. 5a, b) significantly reduces MYCN binding at the promoter regions of *BMAL1* and *RORα* (Supplementary Fig. 5c, d), suggesting that MYCN repression of the clock requires MIZ1.

**RORα activity rescues MYCN-mediated repression of BMAL1 expression and oscillation.** Synthetic ligands for RORs have been recently characterized, including the RORα/γ agonist SR1078[34]. Unlike most family members, the RORs recognize and bind as monomers to specific sequences of DNA (ROR response elements, RORE) in the regulatory region of the target genes. When bound to these elements, they constitutively recruit co-activators resulting in continual transcriptional activation[35]. To test whether RORα signaling is active in NB cells, we used two approaches:

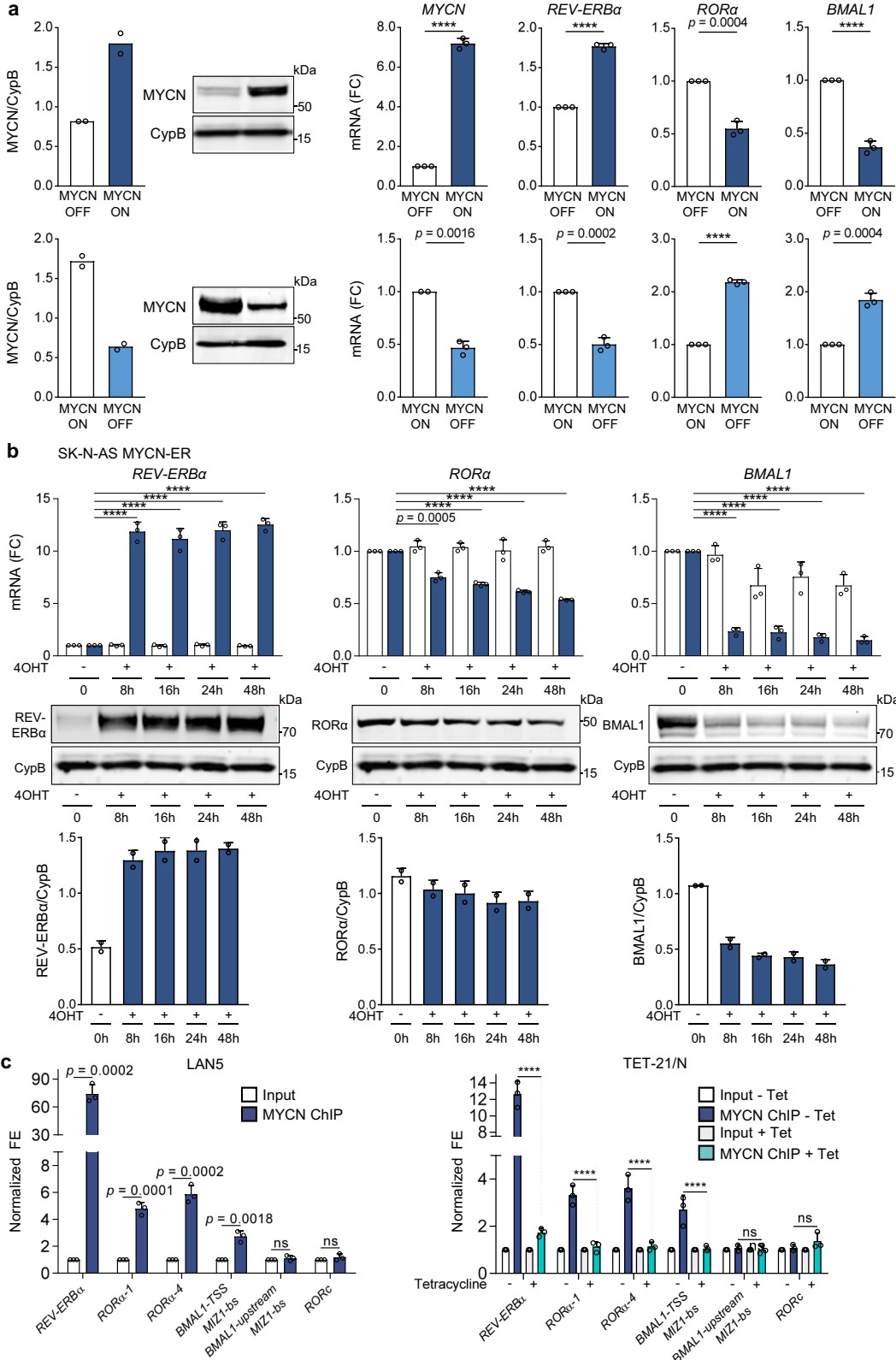

RORα pharmacological activation (via the agonist SR1078) and RORα genetic overexpression. Two MNA cell lines (LAN5 and SK-N-BE(2)-C) were treated with SR1078 and the mRNA expression of RORα target genes containing a RORE in their promoters (*G6Pase, FGF-21*, and *BMAL1*)[36] was determined. SR1078 effectively activates target gene transcription in both cell lines (Fig. 3a). We next examined whether genetic overexpression

of RORα could activate downstream signaling as well. To ectopically express RORα, we inserted a lentiviral DOX-inducible vector containing the sequence of *RORα* (isoform 1) into MNA LAN5 cells because these cells showed relatively low RORα expression compared to other cell lines tested. Conditional RORα overexpression resulted in robust transcriptional activation of its target genes[34] (Fig. 3a). Furthermore, SR1078 and the genetic

**Fig. 2 MYCN directly binds to clock gene promoters and downregulates the positive arm of the clock. a** Left panel: MYCN protein expression in MYCN3 (MYCN Tet-ON) and LAN5 ShMYCN cells after DOX (1 μg/ml) induction for 48 and 72 h, respectively. Protein expression was analyzed by densitometry (Image J v1.42q). CypB served as a loading control and MYCN/CypB ratio was determined ($n = 2$ independent experiments). Right panel: mRNA levels of *MYCN, REV-ERBα, RORα,* and *BMAL1* in MYCN3 (MYCN Tet-ON) and LAN5 ShMYCN cells following DOX induction for 48 and 72 h, respectively. Data are mean ± SD ($n = 3$; ****$p < 0.0001$; two-tailed unpaired *t*-test). **b** mRNA and protein expression of REV-ERBα, RORα, and BMAL1 in SK-N-AS MYCN-ER cells with and without 4OHT (1 μM) for 0, 8, 16, 24, and 48 h. mRNA data are mean ± SD ($n = 3$; ****$p < 0.0001$; two-tailed unpaired *t*-test). Protein expression data were analyzed by densitometry (Image J v1.42q). CypB served as loading control and REV-ERBα/RORα/BMAL1/CypB ratios were determined ($n = 2$). REV-ERBα and RORα were run on the same gel. **c** MYCN ChIP-qPCR assays in LAN5 and Tet-21/N (MYCN Tet-OFF) cells following tetracycline treatment (2 μg/ml for 48 h). Input (white bars) and MYCN IP (blue bars) samples were analyzed by q-PCR using specific primers for *REV-ERBα, RORα, BMAL1,* and *RORc* (Supplementary Table 3). Data are mean ± SD ($n = 3$; ****$p < 0.0001$; two-way ANOVA with Dunnett's multiple comparisons test). FC fold change, FE fold enrichment.

overexpression of RORα efficiently induced BMAL1 protein expression in both systems (Fig. 3b).

We then asked whether SR1078 and RORα overexpression could rescue MYCN-mediated repression of BMAL1 expression and oscillation. For this purpose, we employed two inducible MYCN cell systems: MYCN3 (Tet-ON) and SK-N-AS MYCN-ER cells. As expected, all known and identified MYCN-target genes (*ODC1, REV-ERBα, DKC1, RORα, and BMAL1*) were significantly induced or repressed upon turning on MYCN. Notably, BMAL1 mRNA and protein levels were repressed in both systems upon MYCN activation. However, SR1078 treatment and RORα genetic overexpression were both able to completely rescue BMAL1 mRNA and protein levels (Fig. 3c). Moreover, active MYCN-ER (MYCN-ON) greatly impaired circadian oscillation of *BMAL1* in SK-N-AS MYCN-ER cells. However, overexpression of RORα was capable of both restoring *BMAL1* expression, and also fully rescuing its circadian oscillation (Fig. 3d). Similarly, turning ON RORα highly enhanced *BMAL1* oscillation in MNA cells (Supplementary Fig. 6). Collectively, these data indicate that RORα activity effectively counteracts MYCN-mediated repression of BMAL1 expression and oscillation.

MYC and other cell cycle regulators have been shown to be clock-regulated[37], suggesting that multiple feedback loops exist between MYC and the components of the molecular clock. Thus, we asked whether restoration of the clock could alter c-MYC and MYCN expression. MYCN protein stabilization was notably decreased by ROR agonist SR1078 in MNA cells (LAN5 and SK-N-BE(2)C); however, no changes in c-MYC protein levels were detected in non-MNA c-MYC overexpressing cells (SH-SY5Y) (Supplementary Fig. 7), suggesting that this clock-mediated regulation is MYCN-dependent.

**Restoration of BMAL1 via RORα activation inhibits cell growth and lipogenic gene expression.** Because MYCN repression of the clock correlates with poor clinical outcomes, we then asked whether restoration of the clock via activation of RORα could have a functional significance and alter NB cell phenotype. SR1078 inhibits cell growth in both MNA and non-MNA cell lines. However, this effect is greater in MNA compared to non-MNA cells. Similarly, MNA cells show higher caspase-mediated apoptosis following SR1078 treatment (Fig. 4a). Genetic overexpression of RORα (RORα Tet-ON) also profoundly restricts cell growth and induces cell cycle arrest (Fig. 4b and Supplementary Fig. 8). However, depletion of BMAL1 (Supplementary Fig. 9) promotes cell growth and almost completely rescues the block in cell viability and the apoptosis induced both by SR1078 and RORα overexpression (Fig. 4c), suggesting that the anti-tumor activity of RORα is dependent on BMAL1. Because cell viability assays such as MTT and CCK-8 sense metabolic processes, we used cell counting as an alternative method and confirmed the same changes in cell proliferation (Supplementary Fig. 10). In addition, because SR1078 can stabilize p53[24], we tested whether

this also occurs in NB. We detected no changes in p53 protein levels upon SR1078 treatment in multiple MNA lines. Moreover, genetic depletion of p53 did not affect the response to SR1078 (Supplementary Fig. 11), suggesting that SR1078 alters cell viability in a p53-independent manner.

To further elucidate the molecular mechanisms by which SR1078 impairs cell survival, we performed RNA-sequencing analysis in LAN5 cells with and without SR1078 treatment for 8 h. A total of 712 genes were differentially regulated ($p < 0.05$): 459 downregulated and 253 upregulated (Fig. 4d). Reactome pathway enrichment analysis showed that the most downregulated biological processes were cholesterol biosynthesis and regulation by SREBP (adjusted $p = 2.44\text{E}^{-09}$), metabolism of lipids and lipoproteins (adjusted $p = 1.01\text{E}^{-03}$), and tubulin folding (adjusted $p = 1.39\text{E}^{-03}$), while amino acid transporters (adjusted $p = 9.13\text{E}^{-04}$) were the most upregulated (Fig. 4d and Supplementary Data 1), suggesting that restoration of the clock alters cell metabolism. Overall, 34 out of the 459 SR1078-downregulated genes belonged to the top enriched lipid pathways (cholesterol biosynthesis and regulation by SREBP, and metabolism of lipids and lipoproteins, in blue, Supplementary Fig. 12a). We further selected ten genes specifically involved in cholesterol synthesis (*IDI-1, HMGR, HMGCS1,* and *MVK2*) and FA metabolism (*FABP3, FABP5, ELOVL2, ELOVL6, ACSL3,* and *SCD1*) and validated their expression after SR1078 treatment by q-PCR in MNA cells. Both these gene sets were significantly downregulated by SR1078 in LAN5 cells (Supplementary Fig. 12b). Cell metabolism is tightly controlled by the molecular clock[38] as both metabolites and metabolic gene expression are subject to circadian oscillations[39]. Altogether, these data indicate that activation of RORα inhibits NB cell survival by restoring BMAL1 expression and constraining lipogenic gene expression.

**Activation of RORα restores BMAL1 and blocks NB tumor growth by inhibiting lipid metabolism.** Our patient and in vitro data both suggest that clock activators RORα and BMAL1 are downregulated in MNA NB. These data also suggest that activation of RORα restricts cell survival. We, therefore, examined the in vivo anti-tumor activity of restoring RORα function via SR1078 and genetic overexpression. For this purpose, we used an orthotopic NB mouse model and generated MNA xenografts by implanting MNA cells under the renal capsule of nude mice. This model faithfully recapitulates the aggressive and highly vascular phenotype of primary NB[40]. LAN5 cells were engrafted and SR1078 treatment was initiated 2 weeks after implantation. Vehicle control and SR1078 (15 mg/kg) were given i.p. daily for 14 days, and the effect on tumor growth was compared between groups at the end of the treatment. SR1078 treatment resulted in significant inhibition of tumor growth ($p = 0.02$). The ability of SR1078 to block tumor growth was confirmed in a second MNA xenograft model, derived from MNA IMR32 cells ($p = 0.0004$). Notably, SR1078 anti-tumor activity was associated with the

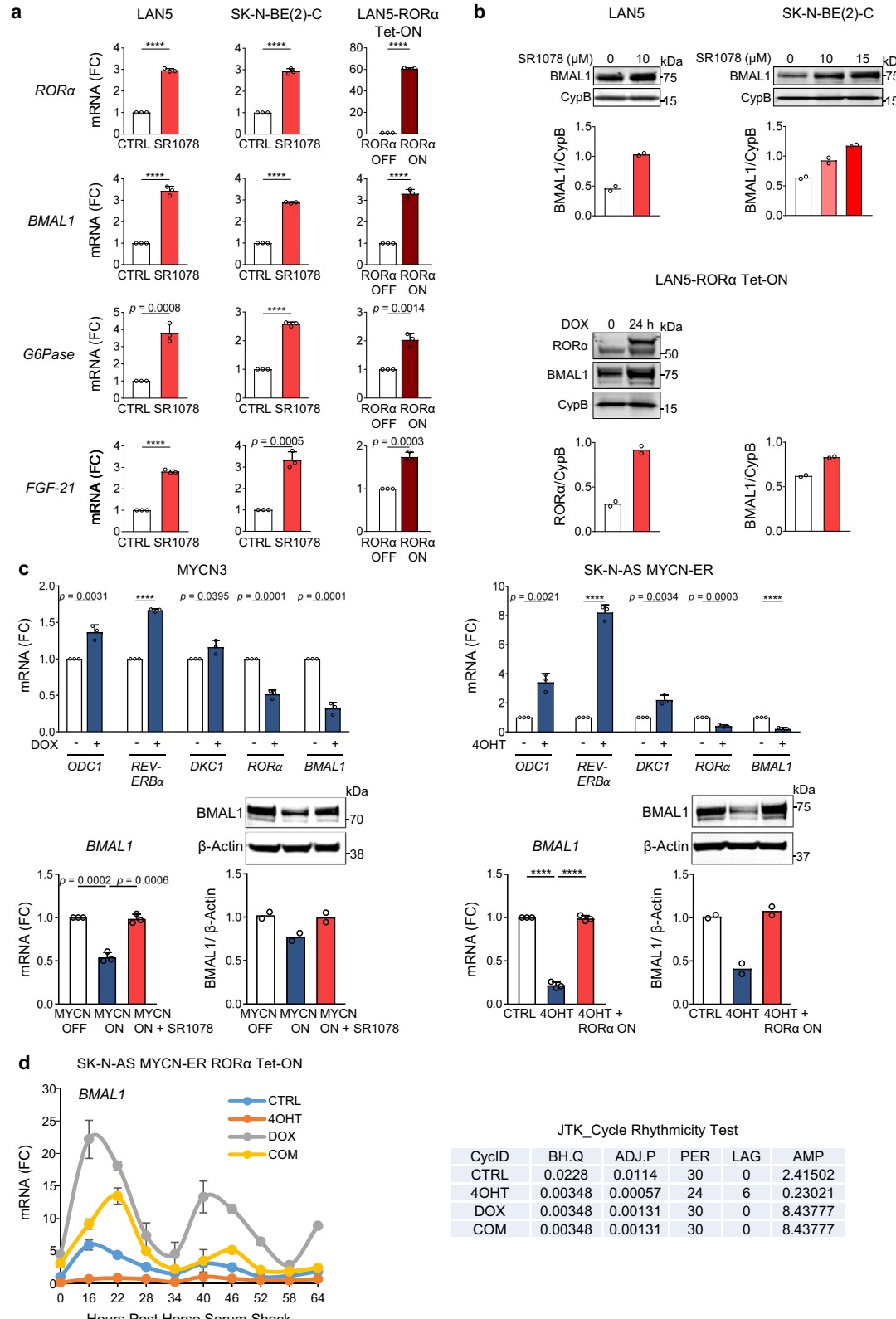

restoration of BMAL1 protein levels in the xenograft tumors (Fig. 5a). However, SR1078 was not able to block tumor growth in a non-MNA xenograft model of SK-N-AS cells ($p = 0.5116$) (Fig. 5b), suggesting that the tumor-suppressive function of RORα activation is MYCN-dependent. The general health conditions of the mice and their body weights were recorded weekly throughout the studies and no apparent signs of toxicity related to

SR1078 treatment were noted (Supplementary Fig. 13). We then engrafted LAN5 RORα Tet-ON cells and evaluated their ability to grow in nude mice with and without DOX treatment to induce RORα overexpression. Overexpression of RORα significantly blocked tumor growth ($p = 0.013$), and DOX-induced tumors also showed higher BMAL1 protein levels (Fig. 5c). In addition, compared to control tumors, tumor cell proliferation was

**Fig. 3 RORα activity rescues MYCN-mediated repression of BMAL1 expression and oscillation. a** mRNA expression of RORα target genes (*BMAL1*, *G6Pase*, and *FGF-21*) in MNA LAN5 and SK-N-BE(2)-C cells treated with SR1078 (10 μM for 24 h), and LAN5 RORα Tet-ON cells cultured with (RORα-ON) and without (RORα-OFF) DOX (2 μg/ml) for 48 h. Data are mean ± SD ($n = 3$; ****$p < 0.0001$; two-tailed unpaired t-test). **b** Upper panel: BMAL1 protein expression in LAN5 and SK-N-BE(2)-C cells treated with SR1078. Lower panel: RORα and BMAL1 protein expression in LAN5 RORα Tet-ON cells cultured with (RORα-ON) and without (RORα-OFF) DOX (2 μg/ml) for 24 h. Data were analyzed by densitometry (Image J v1.42). CypB served as a loading control and BMAL1/RORα/CypB ratios were determined ($n = 2$). **c** Upper panel: mRNA expression of MYCN target genes in MYCN3 cells treated with (MYCN-ON) and without (MYCN-OFF) DOX (1 μg/ml) for 48 h, and in SK-N-AS MYCN-ER cells treated with and without 4OHT (1 μM) for 48 h. Data are mean ± SD ($n = 3$; ****$p < 0.0001$; two-tailed unpaired t-test). Lower panel: BMAL1 mRNA and protein expression in MYCN3 cells under MYCN-OFF, MYCN-ON, and MYCN-ON + SR1078 (5 μM for 24 h) conditions. BMAL1 mRNA and protein expression in SK-N-AS MYCN-ER RORα Tet-ON cells upon MYCN activation (4OHT 1 μM for 24 h) with and without RORα overexpression (DOX 2 μg/ml for 24 h). Protein expression data were analyzed by densitometry (Image J v1.42q) and BMAL1/CypB ratios determined ($n = 2$). **d** SK-N-AS MYCN-ER RORα cells were cultured with and without 4OHT (MYCN-ON/OFF), DOX (RORα-ON/OFF), and 4OHT + DOX for 24 h and then synchronized with 50% horse serum for 2 h. Cells were collected every 6 h from 16 to 64 h after synchronization. 4-OHT and DOX treatments were applied during and after cell synchronization. *BMAL1* mRNA expression was determined by q-PCR and normalized to *18 S* expression. Data are mean ± SEM ($n = 2$, $p < 0.0001$ at all conditions and time points by two-way ANOVA with Tukey's multiple comparisons test). BMAL1 oscillates at all conditions ($p < 0.05$) according to the JTK-rhythmicity test. FC fold change.

significantly lower in tumors that were both treated with SR1078 and overexpressing RORα ($p < 0.0001$ and $p = 0.0012$, respectively). Although SR1078 induced a robust apoptotic response in the treated tumors compared to controls ($p = 0.0358$), only a mild effect on cell apoptosis was observed in RORα-overexpressing tumors (Supplementary Fig. 14). Collectively, our data suggest that restoration of BMAL1 is tumor suppressive in NB.

The RNA-seq data indicated that SR1078 inhibits NB cell survival in part by suppressing genes involved in lipid metabolism (Fig. 4d). To determine whether SR1078 suppresses tumor growth by perturbing lipid metabolism, we performed lipidomics analysis in control ($n = 10$) and SR1078-treated ($n = 8$) LAN5 tumors. Supporting our RNA-seq data, we found that intratumoral levels of glycerolipids, such as triglycerides (TGs) and their precursors diglycerides (DGs) were significantly reduced by SR1078 (FDR < 0.25; Fig. 5d and Supplementary Data 2), suggesting that activation of RORα effectively inhibits in vivo lipogenesis. In contrast, the intratumoral cholesterol esters were increased by SR1078 (Supplementary Fig. 15). This finding challenges our RNA-seq data in which cholesterol biosynthesis genes were suppressed by SR1078, suggesting a potential compensatory upregulation of cholesterol uptake in these tumors. Because FAs are required for DG and TG synthesis, we further determined what changes in FA composition were induced by SR1078 using liquid chromatography-mass spectrometry (LC-MS)-based FA profiling. Notably, SR1078 significantly reduces the levels of intratumoral FAs including C12:0, C16:0, C18:1, and C20:3, all of which serve as side chains of the reduced DG and TG groups (Fig. 5e and Supplementary Table 2). Altogether, these data suggest that SR1078 blocks tumor growth by inhibiting FA metabolism and glycerolipid synthesis.

**Activation of RORα opposes MYCN-driven lipogenesis.** Recent reports have emphasized the role of MYC in reprogramming lipid metabolism to enhance tumorigenesis[41]. MYC induces SREBP1c-dependent de novo FA synthesis and directly activates the transcription of key lipogenic enzymes to support cancer growth[42]. On the other hand, RORs are also key regulators of lipid metabolism with tissue-dependent effects[35]. Nobiletin (a natural flavone)-mediated activation of RORα/γ protects against metabolic syndrome, suggesting a role in metabolic regulation[43]. Moreover, our RNA-seq and in vivo lipidomics data suggest that SR1078 acts by inhibiting cell metabolic processes, in particular lipid biosynthesis. Unlike normal cells, cancer cells highly depend on de novo lipogenesis[44]. We first asked whether the main lipogenic genes (Acetyl-CoA Carboxylase Alpha (*ACACA*), FA synthase (*FASN*), and stearoyl-CoA desaturase 1 (*SCD1*)) behave as direct MYCN targets. ChIP-qPCR analyses confirmed a significant

enrichment of MYCN binding at the promoter regions of all lipogenic genes (Fig. 6a and Supplementary Fig. 16). We then tested whether activation of RORα could repress MYCN-driven lipogenic gene expression. The mRNA levels of the lipogenic genes were all significantly repressed following SR1078 treatment. However, depletion of BMAL1 completely reestablished their levels, suggesting that SR1078 inhibition of lipogenic gene expression is mediated by BMAL1 (Fig. 6b). This observation was further supported by BMAL1 ChIP-qPCR analysis in SK-N-AS MYCN-ER RORα cells, which confirmed reduced BMAL1 binding at the promoters of all three lipogenic enzymes upon activation of MYCN and increased binding upon overexpression of RORα (Fig. 6c), suggesting that MYCN and BMAL1 may compete for binding at these loci. Moreover, SR1078 treatment rescued the induction of lipogenic gene expression mediated by MYCN in MYCN3 (Tet-ON) cells (Fig. 6d).

To further determine the effects of SR1078 on MYCN-induced de novo lipogenesis and desaturation activity, stable isotope labeling and gas chromatography-mass spectrometry (GC-MS) were employed for FA determination. Turning on MYCN expression significantly increases the concentration of de novo synthesized C14:0 (myristic acid) and C16:0 (palmitic acid), and enhances SCD1 activity as indicated by the $^{13}C_{16}$-labeled C16:1 (palmitoleic acid) to $^{13}C_{16}$-labeled C16:0 ratio. However, SR1078 restores these FA levels, suggesting that activation of RORα efficiently blocks MYCN-mediated de novo FA synthesis and desaturation (Fig. 6e). Because the most profound effect of SR1078 was on SCD1 expression and activity, we next examined how *SCD1* expression relates to NB clinical outcome. *SCD1* expression is strongly associated with poor clinical outcomes and MYCN amplification (Fig. 6f). Moreover, ChIP-qPCR analyses in MYCN Tet-OFF cells confirmed a significant enrichment of MYCN binding at the *SCD1* promoter, which is fully abrogated upon turning off MYCN (Fig. 6g), suggesting that MYCN directly transcriptionally induces SCD1 to promote lipogenesis. Selected FAs (e.g., oleic acid), the main product of SCD1, have been shown to rescue cell viability in cancer cells treated with FA synthesis inhibitors (e.g., TOFA)[36]. We found that culture media supplemented with oleic acid partially rescues the viability of MNA cells treated with SR1078 (Fig. 6h), suggesting that SR1078 targets MYCN-mediated SCD1 activity.

**Activation of RORα sensitizes NB tumors to conventional chemotherapy.** We hypothesized that activation of RORα, by restoring the molecular clock and cell metabolism, could sensitize NB to conventional therapy. To investigate whether SR1078 alters the sensitivity of NB cells to conventional therapies such as

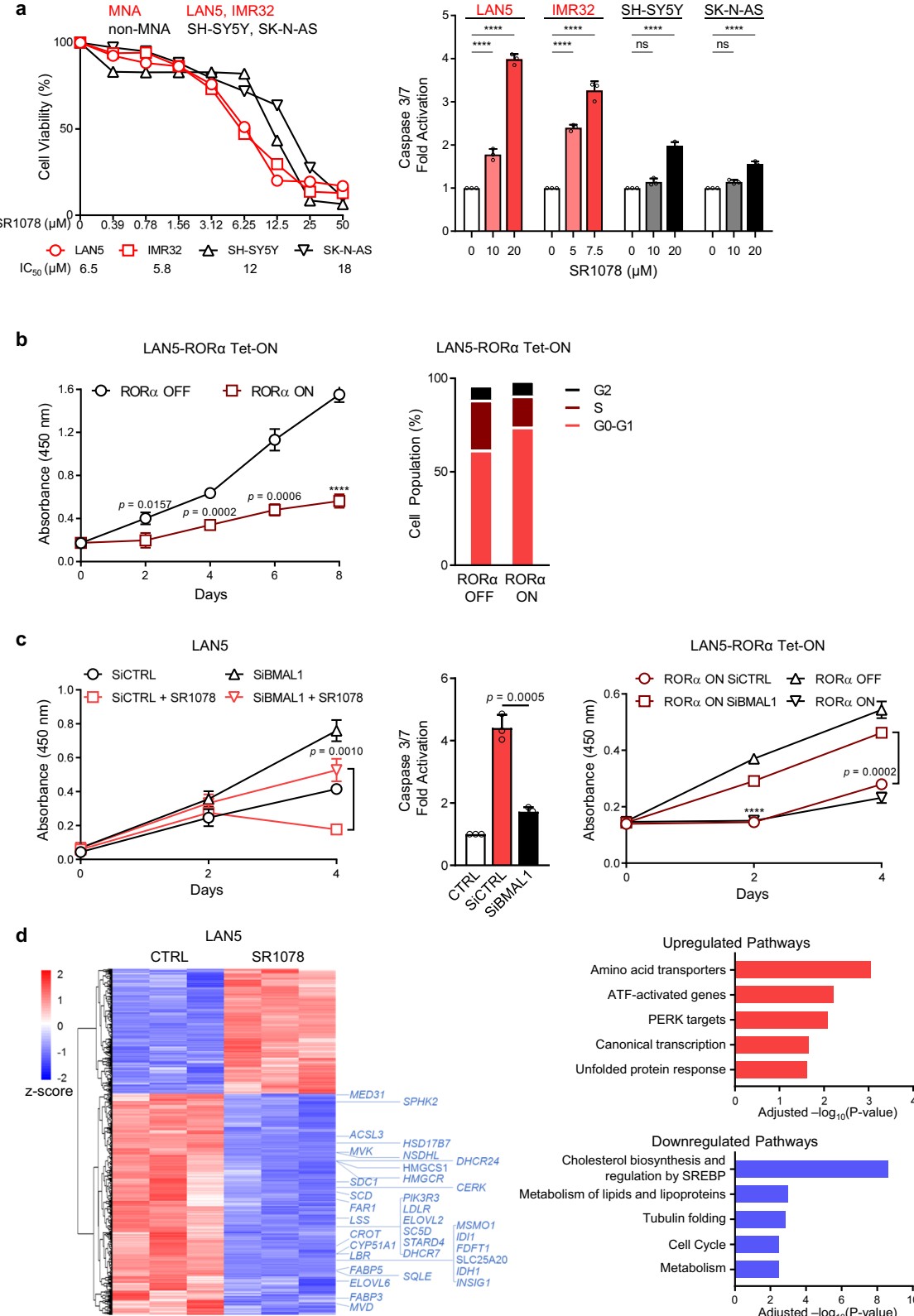

etoposide (VP16), MNA cells were exposed to different concentrations of SR1078 and VP16 as single agents and in combination (IC$_{50}$ doses). VP16 IC$_{50}$ values were significantly reduced (10.3, 4.2, and 10.3 fold for LAN5, IMR32, and NGP cells, respectively) when cells were exposed to the combination therapy (Fig. 7a). SR1078 IC$_{50}$ values were also robustly reduced (13.6, 29, and 40 fold, respectively) when cells were treated with the

combination therapy (Supplementary Fig. 17). Moreover, the addition of low-dose SR1078 strongly enhanced the cell apoptosis induced by VP16 in all three cell lines (Fig. 7a). We then tested the anti-tumor activity of our combination therapy in vivo, using orthotopic xenografts generated from MNA NGP cells. As expected, single agents SR1078 and VP16 significantly blocked tumor growth ($p < 0.05$), and SR1078 strongly sensitized NB

**Fig. 4 RORα activation inhibits cell survival via BMAL1 and constrains lipogenic gene expression. a** Cell viability (MTT assay) and apoptosis (caspase 3/7 assay) in MNA (LAN5, IMR32) and non-MNA (SH-SY5Y, SK-N-AS) cells following treatment with SR1078 for 24 h (apoptosis) and 72 h (cell viability). $IC_{50}$ values are shown in the lower panel. Cell viability data are mean ± SD and are representative from $n = 2$ biological replicates ($n = 4$ technical replicates). MNA cell lines in red; non-MNA cell lines in black. Apoptosis data are mean ± SD ($n = 3$; two-way ANOVA with Tukey's multiple comparisons test). **b** Cell proliferation of LAN5 RORα Tet-ON cells cultured with (RORα-ON) and without (RORα-OFF) DOX over 8 days. Data are mean ± SD ($n = 3$; ****$p < 0.0001$; two-tailed unpaired $t$-test). Cell cycle analysis of LAN5 RORα-OFF and LAN5 RORα-ON cells following 10 days of DOX (2 μg/ml) induction. Data are mean ± SD ($n = 2$). **c** Left panel: cell growth of LAN5 SiCTRL and LAN5 SiBMAL1 cells in the presence and absence of SR1078 (5 μM) for 4 days. Data are mean ± SD ($n = 3$; two-tailed unpaired $t$-test). Middle panel: Caspase 3/7 activation in LAN5 SiCTRL and LAN5 SiBMAL1 cells following SR1078 treatment (10 μM for 24 h). Data are mean ± SD ($n = 3$; two-tailed unpaired $t$-test). Right panel: cell growth of LAN5 RORα SiCTRL and LAN5 RORα SiBMAL1 cells treated with (+Tet, RORα-ON) and without (−Tet, RORα-OFF) DOX for 4 days. Data are mean ± SD ($n = 3$, ****$p < 0.0001$; two-tailed unpaired $t$-test). **d** Left panel: Heat map of differentially expressed genes in LAN5 cells following SR1078 treatment (10 μM for 8 h). Right panel: Most enriched up- and downregulated pathways by SR1078 in LAN5 cells ($p < 0.05$). Blue = downregulated genes, red = upregulated genes.

tumors to VP16 ($p = 0.0031$) (Fig. 7b), suggesting that modulation of the clock could be a viable therapeutic approach.

## Discussion

We have shown here that MYCN suppresses the molecular clock in NB. In turn, restoration of the clock effectively counteracts MYCN-mediated cell growth and metabolism. MYCN-amplification, which occurs in 50% of high-risk NB, distinguishes tumors with a high relapse rate and poor survival[2,45]. We found that MYCN directly activates the clock repressor REV-ERBα while suppressing the activators RORα and BMAL1. This dysregulation is a powerful NB prognostic marker: low *RORα* and *BMAL1* expression and high *REV-ERBα* expression strongly associated with poor clinical outcomes, independently of other prognostic factors.

The downregulation of the clock by MYCN is consistent with previous studies showing that ectopic MYC expression activates REV-ERBα to dampen BMAL1 expression and oscillation[46] and downregulates core clock gene expression via MIZ1[19]. Indeed, in the context of transcriptional repression MYC does not directly contact DNA but associates with DNA-bound transcription factors, such as MIZ1 and/or Sp1[47]. Our data show that depletion of MIZ1 reduces the MYCN binding at the promoter regions of *BMAL1* and *RORα*, suggesting that MIZ1 mediates MYCN repression of the clock in NB cells. This is further supported by the observation that MYCN physically binds the *BMAL1* promoter despite the absence of a functional E-box and in the region where MIZ1 binds. Importantly, we show that MYCN suppresses the clock activators RORα and BMAL1 to sustain cell proliferation and rewire cell metabolism, suggesting that this dysregulation has a functional significance and may contribute to NB oncogenesis. Depending on the context, BMAL1 can function as a tumor suppressor[14] or oncogene[48]. Interestingly, MYC and BMAL1 are basic helix-loop-helix DNA-binding transcription factors, which recognize the same E-box sequences in the regulatory regions of target genes[49,50]. Our ChIP-qPCR data revealed reduced BMAL1 occupancy of target E-box sites in lipogenic genes, which is rescued by ectopic RORα. Thus, we can speculate that oncogenic MYCN plays a role in disrupting BMAL1-controlled transcription and metabolism. *RORα* and *BMAL1* gene silencing is found in several other cancers, including hematological malignancies[26]. RORs also play important roles in tumor metabolism and inflammation. Genetic studies have shown that attenuated RORα leads to various metabolic abnormalities in mice, including hyperglycemia and glucose intolerance[51]. Moreover, RORα inhibits inflammation by interfering with NF-kB signaling[52].

These observations prompted us to investigate whether restoring the clock could counteract the effects of MYCN on cell growth and metabolism. To restore the core clock component BMAL1 and clock function, we pharmacologically activated the circadian activator RORα via SR1078, a synthetic RORα/γ ligand agonist that selectively binds to the ligand-binding domain of the receptors promoting the transcription of target genes, such as *BMAL1*[34]. Our data indicate that restoration of the molecular clock is tumor suppressive in NB. Both SR1078 and genetic overexpression of RORα effectively restore BMAL1 expression and oscillation, and block tumor growth in MNA, but not in non-MNA orthotopic xenografts, suggesting that the anti-tumor effect of SR1078 is MYCN-dependent. Importantly, the anti-tumor activity of SR1078 and RORα is abolished following depletion of BMAL1, further supporting that the core clock component BMAL1 functions as a tumor suppressor in NB. These emerging connections between clock disruption and oncogenesis suggest that circadian interventions could be employed for anti-cancer treatment. Supporting this concept, recent work suggests that pharmacological agonists of REV-ERBs are selectively lethal to cancer cells and block glioblastoma (GBM) and GBM stem cell growth[30,48]. We further hypothesized that SR1078, by restoring the molecular clock and cell metabolism, could be used in combination with conventional therapies to reduce toxicity and enhance chemotherapeutic viability. We find that SR1078 significantly improves the anti-tumor activity of the standard-of-care drug etoposide in MNA xenografts, suggesting that pharmacological modulation of the clock could be explored in future combination studies.

Metabolic rewiring is a key function of MYC oncogenes. Although several c-MYC metabolic functions are well characterized, the role of MYCN in metabolic reprogramming is less defined. MYC, in coordination with SREBP1, controls lipogenesis, which is required for both initiation and maintenance of tumorigenic growth[53]. Moreover, inhibition of FA oxidation dampens tumor growth of triple-negative breast cancers overexpressing MYC[54]. Interestingly, the activity of some of these lipogenic enzymes has been previously reported to be clock-controlled[55] and linked to oncogenesis[56], suggesting that changes in their circadian activity may affect tumor growth. We demonstrate here that MYCN directly induces lipogenic gene expression and lipid intermediates, which are constrained by SR1078 in a BMAL1-dependent fashion. Moreover, SR1078 profoundly inhibits lipid metabolism in vivo by reducing glycerolipids and their FA components. Thus, we postulate that MYCN-mediated repression of the clock could induce the unconstrained expression of lipogenic genes and activate lipid metabolism to support tumor growth. However, restoration of the clock effectively constrains MYCN-mediated lipogenesis. We show that MYCN directly binds and upregulates the stearoyl-CoA desaturase-1 (SCD1), which catalyzes the synthesis of monounsaturated from saturated FA. In turn, activation of RORα blocks MYCN-induced SCD1 expression and activity. Moreover, supplementation with oleic acid, the major product of SCD1, rescues the viability of NB cells, confirming that SR1078 functions by blocking

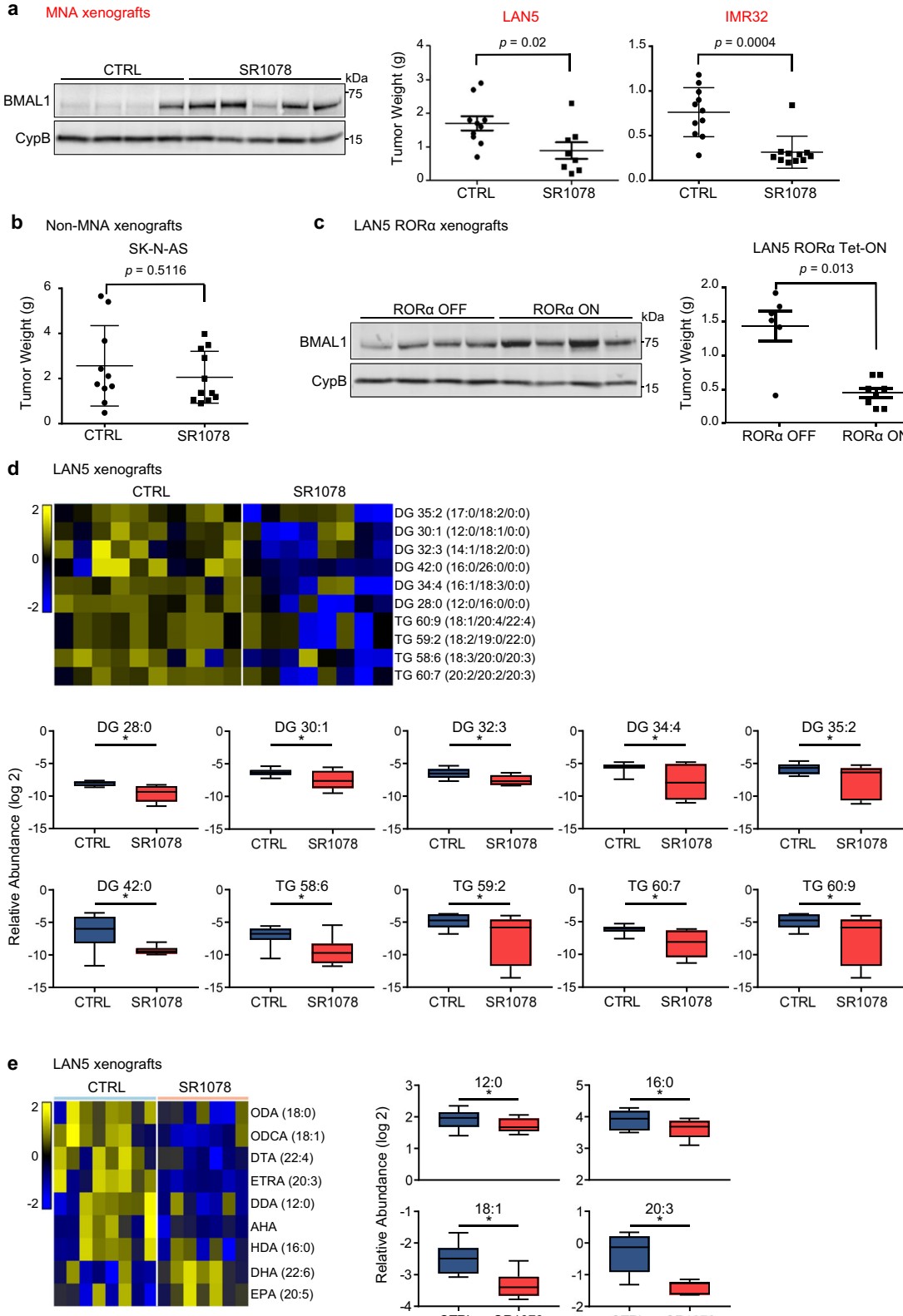

MYCN-mediated lipogenesis. SCD1 expression is frequently elevated in cancer[57] and its inhibition has shown some anti-tumor activity[58]. We find that high *SCD1* expression predicts poor clinical outcomes in NB and its expression directly correlates with MYCN amplification. This could represent a potentially targetable metabolic vulnerability.

Overall, our results demonstrated that restoration of the molecular clock via activation of RORα blocks MYCN-driven tumor growth and lipid metabolism (Supplementary Fig. 18). These novel findings implicate RORα activation as a potential therapeutic strategy for blocking MYCN-mediated dysregulation of the molecular clock and metabolism in high-risk NB.

**Fig. 5 SR1078 and RORα genetic overexpression block tumor growth and inhibit lipid metabolism. a** Tumor weights (mean ± SD) from MNA LAN5- and IMR32-derived xenografts treated with vehicle control or SR1078 (15 mg/kg i.p. for 14 days) (LAN5: $p = 0.02$, Mann–Whitney test, control group $n = 10$ and SR1078 group $n = 8$; IMR32: $p = 0.0004$, Mann-Whitney test, control group $n = 11$ and SR1078 group $n = 11$). BMAL1 protein expression in representative LAN5-xenografted tumor lysates from SR1078 treated mice and controls. **b** Tumor weights (mean ± SD) from non-MNA SK-N-AS derived xenografts treated with vehicle control or SR1078 (15 mg/kg i.p. for 14 days) ($p = 0.5116$, Mann–Whitney test, control group $n = 10$ and SR1078 group $n = 10$). **c** Tumor weights (mean ± SD) from DOX-induced (RORα-ON) and controls (RORα-OFF) mice ($p = 0.013$, Mann–Whitney test, RORα-OFF group $n = 6$ and RORα-ON group $n = 8$). BMAL1 protein expression in tumor lysates from representative RORα-ON and RORα-OFF mice. **d** Heat map and relative levels of altered DG and TG in LAN5-derived xenografts treated with vehicle control ($n = 10$) or SR1078 ($n = 8$; 15 mg/kg i.p. for 14 days). Groups were compared by a two-tailed unpaired *t*-test. *P*-values were adjusted by the Benjamini–Hochberg method to obtain FDR. Changes with FDR < 0.25 were selected for heat map presentation; *=FDR < 0.25. Yellow = upregulated lipids; blue=downregulated lipids. **e** Heat map and relative levels of altered intratumoral FAs (*=FDR < 0.25) in LAN5-derived xenografts treated with vehicle control ($n = 8$) or SR1078 ($n = 7$; 15 mg/kg i.p. for 14 days). Groups were compared by a two-tailed unpaired t-test. P-values were adjusted by the Benjamini-Hochberg procedure to obtain FDR. Changes with FDR < 0.25 were selected for heat map presentation; *=FDR < 0.25. AHA aminoheptanoic acid, DDA dodecanoic acid, DG diglycerides, DHA docosahexaenoic acid, DTA docosatetraenoic acid, EPA eicosapentaenoic acid, ETRA eicosatrienoic acid, HDA hexadecanoic acid, ODA octadecanoic acid (stearic acid), ODCA octadecenoic acid (oleic acid); TG triglycerides. For panels (**d**) and (**e**): the box plot is defined by two lines at the 25th and 75th percentile. A line is drawn inside the box at the 50th percentile.

## Methods

**Cell lines and culture conditions**. SH-SY5Y, IMR32, SK-N-BE(2)-C, SK-N-AS, NGP (ATCC), LAN5, and CHLA255 (Metelitsa, BCM, Houston, TX), and MYCN3 (Shohet, BCM, Houston, TX) NB cell lines were maintained in RPMI 1640 medium. Tet-21/N (Perini, Bologna, Italy) cells were grown in DMEM high glucose medium and kept in selection with neomycin (G418, Santa Cruz Biotechnology# sc-29065a, 0.2 mg/ml) and hygromycin (Sigma Aldrich# H3274, 0.150 mg/ml). All cell lines were validated by genotyping and regularly tested for mycoplasma. To turn off MYCN, doxycycline (DOX; Santa Cruz Biotechnology# sc-204734A) was added at a final concentration of 2 μg/mL. SK-N-AS MYCN-ER (Altman, Rochester, USA) cells were maintained in DMEM high glucose and sodium pyruvate. To activate MYCN transcription, cells were treated with 1 μM 4-hydroxytamoxifen (Sigma Aldrich# T176) for 48 h. Cell lines were validated by genotyping and confirmed by expression of CD56, Nestin, MYCN, and tyrosine hydroxylase within the past 12 months. SR1078 (Calbiochem# 557352) was used at a concentration of 5–15 μM for in vitro studies. BSA-conjugated oleic acid (Sigma Aldrich# O3008) was supplemented in the culture media (10–500 μM).

**Primary NBs survival data**. mRNA gene expression analyses were performed within R2: a genomics analysis and visualization platform (http://r2.amc.nl), using datasets: NB Kocak ($n = 649$, GEO: GSE45547, patient cohort 1), NB Versteeg ($n = 88$, GEO: GSE16476 88/122, patient cohort 2), and NB Research Consortium (NRC; $n = 283$, GEO: GSE85047, patient cohort 3). In the Kocak dataset, 476 of the 649 samples were annotated with survival data. To test the association of gene expression with survival, individual gene expression profiles were dichotomized by a median split into "high" or "low" groups, and Kaplan–Meier survival curves were plotted for each group. A Cox regression model was used to test for the independent predictive ability of RORα expression after adjustment for other significant factors: MYCN amplification, age, and INSS stages. The Mann–Whitney U test was used to evaluate the significant differences among quantitative variables. The significance of correlation analyses between MYCN and RORα was evaluated by the Spearman test.

**Plasmid constructs**. To generate MYCN-inducible MYCN3 cells, *MYCN* cDNA was cloned into a pTR2-Hygro vector (BD Biosciences) containing a tetracycline-responsive promoter. The construct was then transfected into a SHEP sub-clone stably expressing the tetracycline response element and selected with hygromycin[59]. For LAN5 ShMYCN cells, GIPZ human MYCN lentiviral clones (V2LHS_36755 and V3LHS_322662) were obtained from the Cell-Based Screening Service core at Baylor College of Medicine (BCM). MYCN–shRNA sequences were inserted in the lentiviral gene silencing vector pInducer11, which has a dual fluorescent system consisting of a constitutive cassette (rtTA3 and eGFP) to detect infection efficiency and a turboRFP (tRFP)-shRNA cassette activated upon DOX treatment. To generate LAN5 and SK-N-AS RORα conditional overexpressing cells, *RORα* isoform 1 cDNA (Origene# RC202926) was cloned into the pENTR/D-TOPO vector (Invitrogen) and then gateway cloned into a pInducer21 lentiviral vector with RORα-HA under the control of the tetracycline-inducible promoter. LAN5 cells were transduced with lentiviral vectors at a multiplicity of infection sufficient to achieve greater than 90% infection as determined by the presence of GFP fluorescence. BMAL1 siRNAs (scrambled and BMAL1 targeted) were obtained from Thermo Scientific (Dharmacon SMARTpool siRNAs, which consists of a pool of four siRNA sequences, #SO-2564383G). To generate LAN5 Shp53 cells, second-generation lentiviruses expressing Shp53 and ShLuc control were used as previously described[60]. Briefly, 293 T cells were transfected with pLSLPw construct along with pVSVG and pLV-CMV-delta 8.2 by using lipofectamine. Virus-containing supernatants were collected at 48 h and 72 h and NB cells transduced in the presence of 8 mg/ml polybrene (Sigma).

**In vitro functional assays**. To test the effect of SR1078 on cell viability, 3-(4,5-Dimethylthiazol-2-yl)-2,5-diphenyltetrazolium bromide (MTT) assay was carried out as previously described[61]. Cell proliferation was assessed using the CCK-8 Kit (Dojindo, Kumamoto, Japan) according to the manufacturer's instructions. Briefly, $3-5 \times 10^3$ cells/well were seeded in 96-well plates. At determined times, CCK-8 reagent (10 μl) was added to the wells and incubated at 37 °C for 4 h, and absorbance (450 nM) was measured. Caspase 3/7-mediated apoptosis was determined using the Caspase-Glo assay kit (Promega). Briefly, $5 \times 10^3$ cells/well were seeded in 96-well plates and subjected to different treatments for 24–48 h. Equal volumes of caspase reagent were added to the wells, and luminescence was measured in a plate-reading luminometer after 3 h incubation. Cell cycle distribution was measured by PI Flow Cytometry Kit (abcam# 139418). LAN5 RORα cells were harvested and fixed (70% ethanol for 2 h). Cells were stained with 50 μg/ml PI for 30 min before FACS analysis (BD LSRII with BD FACSDIVA v6.1.3). Cell population (%) in G0-G1, S, and G2 phases was analyzed by FlowJo v7.6.1.

**Real-time q-PCR and western blotting**. Total RNA was extracted and purified with RNeasy kit (QIAGEN). Real-time PCR was performed with SYBR green reagent using a 50 ng template in a 25 μl reaction mixture according to the manufacturer's protocol. Levels of GAPDH and CUL1 were used for normalization. All primers were designed with one primer overlapping an exon boundary. Primer sequences are listed in Supplementary Table 3. For western blotting analyses, cells and tumor tissue were lysed with RIPA buffer (Sigma) containing protease inhibitors (Roche). Protein concentration was measured by Bradford assay (Bio-Rad reagent) and 50–75 μg protein was electrophoresed and transferred. Cyclophilin B (CypB) was used as the protein loading control. Primary antibodies: MYCN (Cell Signaling# 9405 S; 1:500 dilution); c-MYC (Abcam# ab32072; 1:500 dilution); RORα (Santa Cruz Biotechnology# sc-6062; 1:500 dilution); p53 (Santa Cruz Biotechnology# sc-126; 1:500 dilution); BMAL1 (Cell Signaling# 14020 S; 1:1000 dilution); REV-ERBα (Cell Signaling# 13418 S; 1:500 dilution) and CypB (Santa Cruz Biotechnology# sc-130626; 1:1000 dilution). Secondary antibodies: anti-mouse IgG IRDye680RD (Licor# 926-68070; 1:10000 dilution) and anti-rabbit IgG IRDye800CW (Licor# 926-3211; 1:10000 dilution). Blots were scanned on an Odyssey infrared imaging system (LI-COR) and analyzed by Odyssey® Application Software v3.0. Protein expression was quantified by densitometry (ImageJ v1.42q). All uncropped blot scans are provided in the Source Data file.

**RNA-seq analysis**. We generated libraries with the Illumina TruSeq Stranded mRNA (p/n 20020594) kit starting with 150 ng of RNA, and we sequenced with the NextSeq 500, PE75, v2.5 reagent kit. Differential expression programs from RNA-seq profiles of LAN5 cells treated with SR1078 (10 μM) for 8 h were identified. Fastq files were aligned to the reference genome (hg38.p12) using STAR (v2.4) with default settings to produce transcripts per million (tpm) estimates for each ensembl gene[62]. When identifying up- and downregulated genes, we required genes to be significantly dysregulated by t-test and with a consistent fold change above 1.5 or below 0.67 compared to controls. Expression values were offset by adding a pseudo count of 1 tpm for each gene to exclude genes with low expression estimates. Gene set enrichment for Reactome pathways based on dysregulated genes was performed in Enrichr[63]. Pathways in the same category were grouped. Significant genes were sorted by p-value and associated with pathway terms.

**Chromatin immunoprecipitation (ChIP)**. The MYCN ChIP assays were performed as follows. Briefly, $1 \times 10^7$ cells were cross-linked using 1% formaldehyde, and the reaction was stopped using 0.125 M Glycine. The cell pellet was resuspended in cell lysis buffer and following 3500×*g* centrifugation, RIPA sonication buffer was added to complete nuclei lysis. DNA shearing was performed by

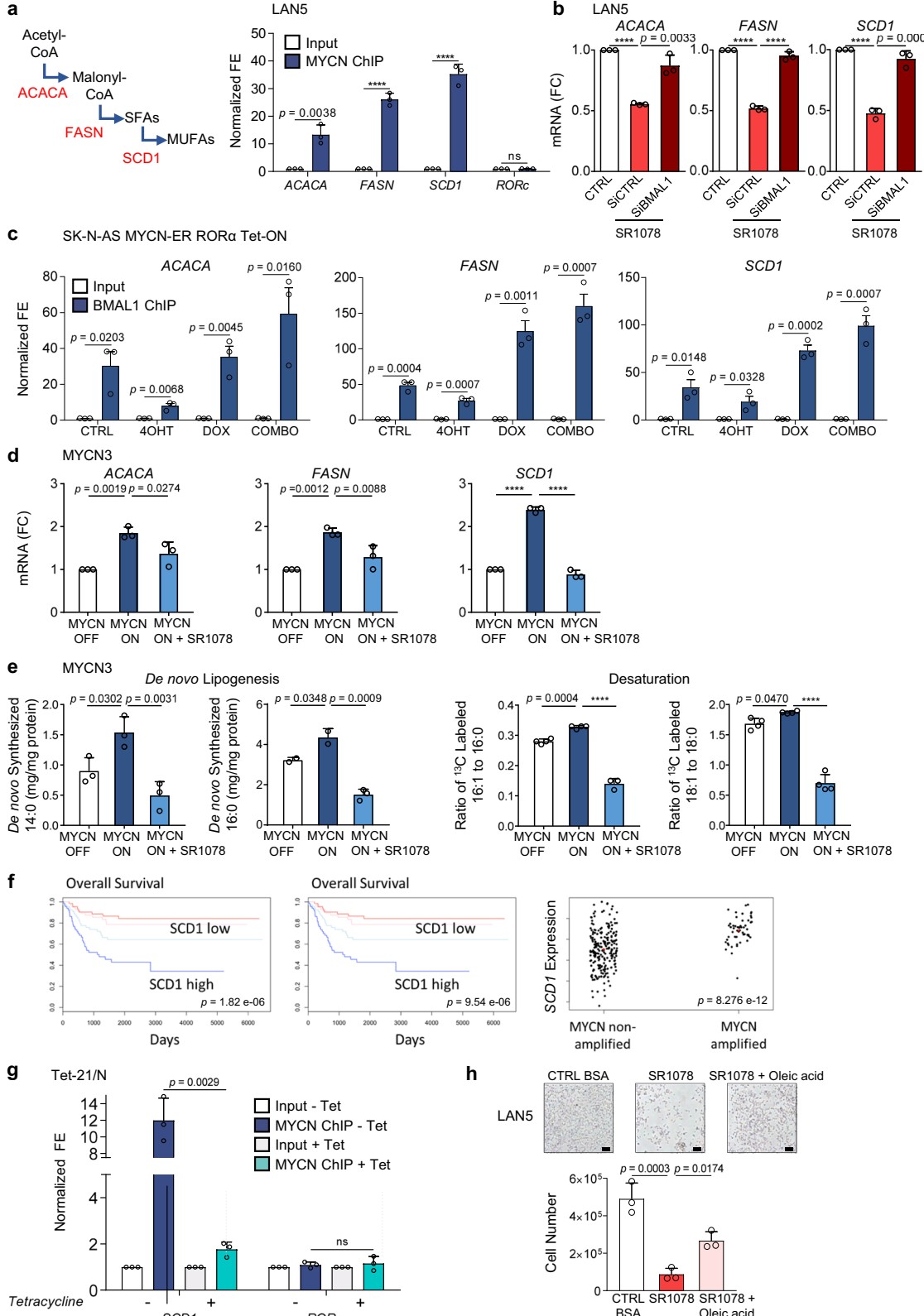

sonication using the Bioruptor PLUS (Diagenode). A small aliquot of sonicated material was put aside, and the remaining sample was immunoprecipitated using 5 μg of ChIP-grade antibodies (B8.4.B, Santa Cruz Biotechnology# sc-53993). Rec-sepharose Protein A or G beads (Invitrogen) were used to immobilize immuno-complexes and RNAse-A treatment (37 °C 1 h) and reverse crosslinking was per-formed using Proteinase K (Roche) for 6 h at 65 °C. Immunoprecipitated DNA was purified using phenol/chloroform and ethanol precipitation techniques. DNA was analyzed by q-PCR and normalized by the fold enrichment method ($2^{-(\Delta\Delta CT)}$)

using the primers listed in Supplementary Table 3. The ABCA10 transcription start site was used as a negative control DNA region for MYCN. The BMAL1-ChIP assays were performed using the ChIP-IT® Express Enzymatic (Active Motif# 53009) following the manufacturer's protocol. Approximately, $1.5 \times 10^7$ cells were fixed using 1% formaldehyde. The fixation reactions were stopped by adding Glycine Fix-Stop solution. After washing with ice-cold PBS, cells were collected in a solution containing PMSF and centrifuged at 4 °C. The cell pellet was disrupted with a Dounce homogenizer in 1 ml ice-cold lysis buffer containing protease

**Fig. 6 SR1078 opposes MYCN-induced lipogenesis. a** Left panel: schematic representation of the main enzymatic steps of de novo lipogenesis. Right panel: MYCN ChIP assays in LAN5 cells (input = white bars; MYCN IP = blue bars). Samples were analyzed by q-PCR using specific primers for lipogenic enzymes (*ACACA*, *FASN*, and *SCD1*). *RORc* served as negative control (Supplementary Table 3). Data are mean ± SD ($n = 3$ biological replicates; ****$p < 0.0001$; two-tailed unpaired *t*-test). **b** mRNA expression of lipogenic enzymes in LAN5 SiCTRL and SiBMAL1 cells upon SR1078 treatment (10 µM for 24 h). Data are mean ± SD ($n = 3$; ****$p < 0.0001$; two-tailed unpaired *t*-test). **c** BMAL1 ChIP-qPCR assays in SK-N-AS MYCN-ER RORα cells following 4OHT (1 µM for 24 h), DOX (2 µM for 24 h), and 4OHT + DOX treatments. Data are mean ± SD ($n = 3$; two-tailed unpaired *t*-test). **d** mRNA expression of lipogenic enzymes in MYCN3 (MYCN-ON) cells upon SR1078 treatment (10 µM for 24 h). Data are mean ± SD ($n = 3$; ****$p < 0.0001$; one-way ANOVA with Dunnett's multiple comparisons test). **e** Determination of de novo lipogenesis and SCD1 activity in MYCN3 (MYCN-ON) cells upon SR1078 treatment (5 µM for 72 h). Data are mean ± SD ($n = 4$ biological replicates; ****$p < 0.0001$; one-way ANOVA with Dunnett's multiple comparisons test). **f** Kaplan–Meier analysis of overall survival (OS) and event-free survival (EFS) in the NB NRC patient cohort ($n = 101$) based on *SCD1* mRNA expression. Significant ($p = 8.276$ E$^{-12}$ by Fisher's exact test) association with MYCN amplification is shown. **g** MYCN ChIP assays in Tet-21/N cells in conditions of MYCN-ON and MYCN-OFF (DOX for 48 h) (input = white bars; MYCN IP = blue bars). Samples were analyzed by q-PCR using specific primers for *SCD1* and *RORc* (Supplementary Table 3). Data are mean ± SD ($n = 3$ biological replicates; two-tailed unpaired *t*-test). **h** Cell viability of LAN5 cells treated with SR1078 (10 µM for 48 h) in control (CTRL) media (0.3 mM BSA) or CTRL media supplemented with oleic acid (500 µM for 48 h). Data are mean ± SD ($n = 3$ biological replicates; one-way ANOVA with Dunnett's multiple comparisons test). Representative images (100×) of LAN5 cells; scale bar = 50 µm. FC fold change, FE fold enrichment.

inhibitor and PMSF. After centrifugation, nuclei were resuspended in digestion buffer containing protease inhibitor and PMSF, then incubated at 37 °C for 5 min. Chromatin was then sheared to 200–300 bp by adding an enzymatic shearing cocktail for 10 min at 37 °C. Samples were centrifuged at 21,000xg for 10 min at 4 °C. One percent aliquot of this material was retained as "input" DNA. The remaining chromatin sample was divided, one-half was immunoprecipitated with the BMAL1 antibody (Abcam# 93806), and the second half was used for a mock immunoprecipitation with a control IgG. Real-time PCR amplification was carried out using 2.5 µl of DNA sample, using primers listed in Supplementary Table 3.

**Circadian oscillation studies**. *Real-time bioluminescence monitoring:* MYCN3 and LAN5 cells were treated with or without 1 µg/ml DOX for 48 h before infection with Per2-dluc adenovirus as previously described[64] for another 24 h. Cells were then serum shocked with 50% horse serum for 30 min before being subjected to real-time bioluminescence as previously described[32]. *BMAL1 circadian oscillation:* SK-N-AS MYCN-ER RORα cells were cultured with and without 4OHT (MYCN-ON/OFF), DOX (RORα-ON/OFF), and 4OHT + DOX for 24 h and then synchronized with 50% horse serum for 2 h. Cells were collected every 6 h from 16 to 64 h after synchronization. 4-OHT and DOX treatments were applied during and after cell synchronization. *BMAL1* mRNA expression was determined by q-PCR and normalized to *18 S* expression. To test for rhythmicity, JTK_Cycle[65] was applied. For serum shock experiments, rhythmicity was determined from all values excluding the ZT0 (unsynchronized) time point.

**FA measurement**. FA tracing was performed using pentafluorobenzyl bromide (PFBBr) derivatization and GC-MS negative chemical ionization as previously described[66]. SP-2380 columns (Supelco, Inc.) were used to allow better peak separation of palmitoleic from sapienic acid. Data were acquired in selective ion monitoring mode. The fragment masses m/z of 227–229 were used to monitor $M_0$–$M_2$ in myristic acid; 255–257 and 271 for $M_0$–$M_2$ and $M_{16}$ in palmitic acid; and 253–255 and 269 for $M_0$–$M_2$ and $M_{16}$ in palmitoleic acid. Analyte and standards peak areas were determined. The ratio of the area from the sum ($M_0$–$M_2$) analyte-derived ions compared to that from the tridecanoic acid (C13:0) (m/z, 213–215) internal standard was calculated. The ratios were then compared with the calibration curves to determine the concentration of each FA. To determine de novo lipogenesis, deuterated water ($D_2O$) (atom 99%, Cambridge Isotope Laboratory) was added to the media to reach an enrichment of 2.5%. The percentage contribution of newly made FA was calculated as total $^2$H-labeled FA/($^2$H-labeled body water × n) × 100 ($n$ = number of exchangeable hydrogens). Total $^2$H-labeled FA was calculated as $EM_1 \times 1 + EM_2 \times 2$, where $E$ is the enrichment of the isotopomer. The absolute amount of newly synthesized FA was determined as the percentage of newly made FA × the concentration of the total FA. The FA amount was then normalized by the protein content of each sample. For desaturase activity measurement, [U-$^{13}$C] palmitic acid (99 atom%$^{13}$C, Cambridge Isotope Laboratories) was added (1.0 mg/100 mL) to the incubation media and $D_2O$ mix. The concentration of [$^{13}C_{16}$] in palmitic or palmitoleic acid was calculated as the area of $M_{16}$/sum area ($M_0$–$M_2$) × the concentration of FA. Desaturase activity was measured as the ratio of [$^{13}C_{16}$] palmitoleic acid to [$^{13}C_{16}$] palmitic.

In vivo, FA profiling was performed using high-performance liquid chromatography (HPLC) coupled to Agilent 6495 QQQ mass spectrometry (Agilent Technologies) using ESI negative ionization via single reaction monitoring. FAs were extracted as previously described[67]. Separation was performed on a Luna 3u phenyl-hexyl column (150 × 2 mm) using 10 mM ammonium acetate pH 8 (mobile phase A) and methanol (mobile phase B). The binary pump flow rate was 0.2 ml/min with a gradient starting at 40% B at 0 min, 50% at 8 min, 67% at 13 min, 100% at 23 min, and 40% at 30 min to the initial condition and re-equilibration till the end of the gradient at 37 min. Data were

analyzed with Agilent mass hunter quantitative software (v10.0) and were normalized with an internal standard and log2 transformed per sample. For every FA in the normalized dataset, two-sample *t*-tests were conducted to compare expression levels between different groups. Differential metabolites were identified by adjusting the p-values for multiple testing at FDR < 0.25.

**In vivo studies**. In vivo studies were approved by the Institutional Animal Care and Use Committee of BCM (AN7089). Orthotopic xenografts of human NB were generated as previously described[40]. Briefly, an inoculum of $10^6$ tumor cells in 0.1 mL of PBS was injected under the renal capsule of 4–6-week-old female athymic Ncr nude mice (Taconic). Two weeks after cell implantation and confirmation of successful tumor engraftment by bioluminescent imaging (Xenogen IVIS 100 System, Caliper Life Sciences), mice were randomized and divided into treatment groups. Four to five weeks post-implantation mice were sacrificed, tumors resected, weighed, and preserved in 4% paraformaldehyde for immunohistochemical analysis. All animal data were compared using the Mann–Whitney test. To induce RORα expression DOX Hyclate (Sigma Aldrich# D9891) was given as an intraperitoneal injection (5 mg/kg) at day −1, 0, and +1 post-implantation and weekly for three weeks. DOX (2 mg/ml) was then supplemented in 5% sucrose drinking water. SR1078 (Calbiochem# 557352) was administered i.p. daily (15 mg–25 mg/kg in DMSO). Etoposide (VP16, Sigma Aldrich# E1383) was administered i.p. three times a week (15 mg/kg in DMSO) for two weeks. Vehicle control (PBS, PEG (5.4%) Tween 20 (5.4%), DMSO (4%)) was administered i.p. daily. Combination therapy of VP16 (10 mg/kg) and SR1078 (25mg/kg) was administered i.p. daily for two weeks. *Immunohistochemistry staining:* Assessment of tumor cell proliferation and apoptosis was performed as previously described[68]. Briefly, paraffin-embedded tumor sections were blocked with horse serum (10%) and incubated with Ki67 antibody (BD Biosciences# 550609; 1:400 dilution) or cleaved caspase 3 antibody (Cell Signaling# 9661 L; 1:400 dilution) at 4 °C overnight. Sections were washed with PBS and incubated with biotinylated anti-mouse (BA# 9200; 1:200 dilution) and anti-rabbit (BA# 1000; 1:200 dilution) antibodies at room temperature for 30 min, following incubation with the 3,3′-diaminobenzidine solution and counterstaining with hematoxylin. For tumor section scoring, four fields were evaluated with 500 tumor cells in each field using light microscopy, and percent positive tumor cells were calculated.

**Lipidomics**. Sample Preparation was as follows: Tumor samples were homogenized with an equal ratio of 0.15 M KCl and methanol followed by 400 µL of dichloromethane and 2 µL of acetic acid. Internal standards and quality control samples were prepared as previously described[69]. Briefly, an equimolar concentration of 10 µL of internal standards was added to each sample before adding water and dichloromethane (1:1). After centrifugation (5 min 6500×g), the lower organic layer was collected and dried under nitrogen. Before mass spectrometric (MS) analysis, the dried extract was suspended in 100 µL of solution (10:5:85 acetonitrile/water/isopropyl alcohol containing 10 mM ammonium acetate) and subjected to liquid chromatography/triple time-of-flight (TOF) for analysis (Shimadzu CTO-20A Nexera X2 41 UHPLC system). MS analysis was carried out on a Triple TOF 5600 equipped with Turbo VT$^M$ ion source (AB Sciex). Data were acquired with Analyst TF software v1.8 (AB Sciex). Lipids were separated on the acquity HSS UPLC T3 column (Waters) at 55 °C. The initial mobile phase consists of acetonitrile/water (40:60 v/v) with 10 mM ammonium acetate, and acetonitrile/water/isopropanol (10:5:85 v/v) with 10 mM ammonium acetate with gradient elution at a flow rate of 0.4 mL/min and an injection volume of 5 µL. Operating source conditions for Triple TOF scan for positive ionization: source voltage 5500 V, declustering potential (DP) 60 V, source T 450 °C, ion source gas 1 (GS1) 40 psi, ion source gas 2 (GS2) 45 psi, curtain gas (CUR) 30 psi, and collision energy 10 V. Conditions for negative ionization: source voltage −4500V, DP 60 V, GS1 40 psi, GS2 45 psi, CUR 30 psi, and collision −10 V. The MS/MS spectra were controlled by data-dependent

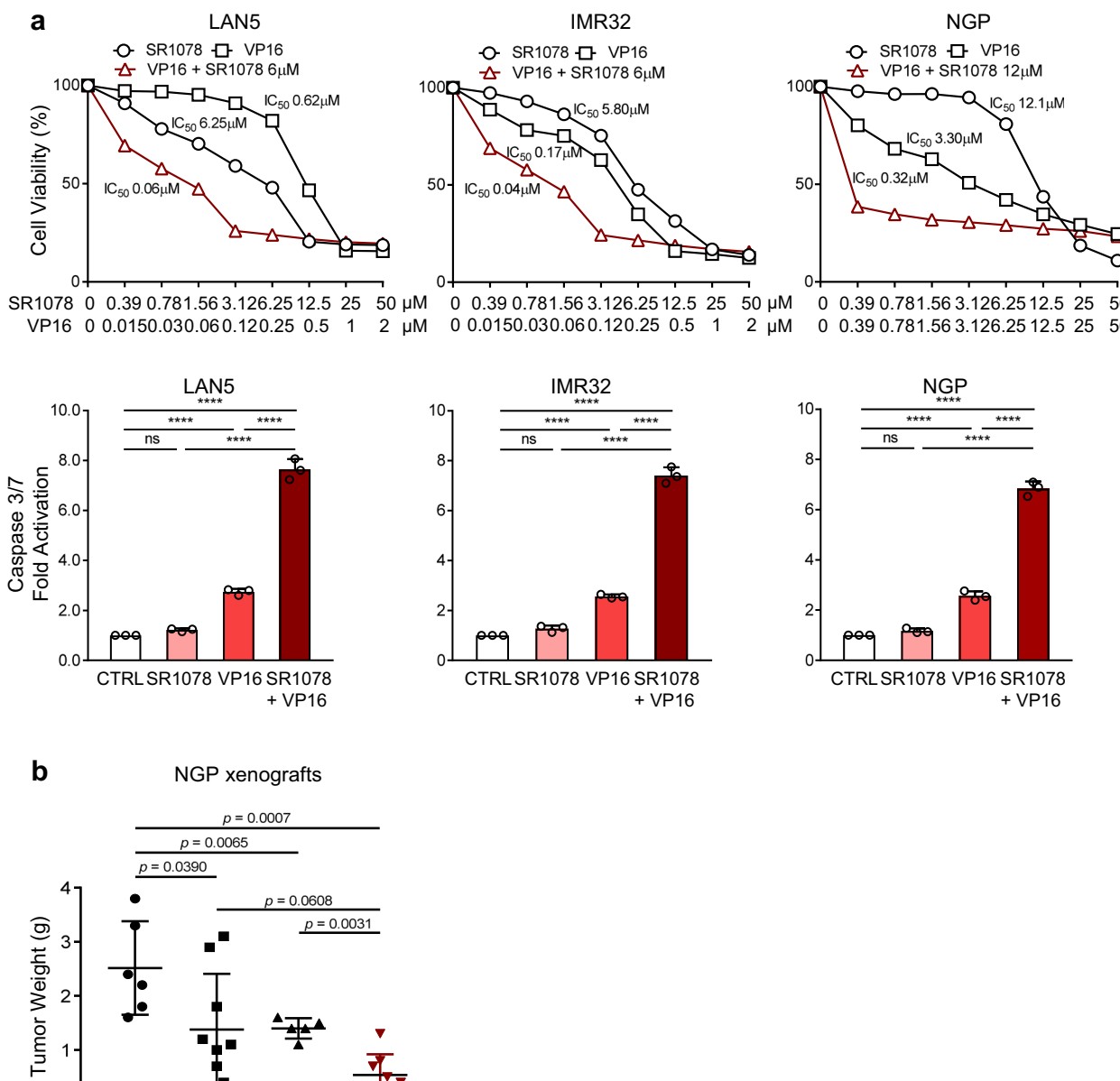

**Fig. 7 SR1078 sensitizes NB to conventional chemotherapy. a** Cell viability (72 h) and caspase 3/7 activation (24 h) of MNA cells (LAN5, IMR32, and NGP) upon treatment with SR1078, etoposide (VP16), and their combination. Cell viability data are means and are representative from $n = 2$ biological replicates ($n = 4$ technical replicates). Apoptosis data (SR1078 2.5 μM; VP16 100–200 nM) are means ± SD ($n = 3$; ****$p < 0.0001$; one-way ANOVA with Tukey's multiple comparisons test). **b** Tumor weights (mean ± SD) from NGP-derived xenografts treated with vehicle control, SR1078 (20 mg/kg i.p. daily), VP16 (10 mg/kg i.p. 3 times/week), and their combination (in red). All treatments continued for 14 days (Mann–Whitney test; control group $n = 6$; SR1078 group $n = 9$; VP16 group $n = 5$; combination group $n = 8$).

acquisition with dynamic background subtraction, charge monitoring, and dynamic exclusion of former target ions for 9 s. Rolling collision energy spread was set to generate a collision energy ramp around a collision energy center point with the goal of providing a richer MS/MS spectrum of lipids. Mass accuracy was maintained by the use of an automated calibrant delivery system interfaced to the second inlet of the DuoSpray source. Data processing and statistical analysis were performed as previously described[67]. Briefly, data were normalized by median inter-quantile range normalization and were log$_2$ transformed. The two groups were compared by Student's $t$-test. $P$-values were adjusted by the Benjamini–Hochberg procedure to obtain FDR. Changes with FDR < 0.25 were selected for heat map presentation.

**Statistical analysis.** Data were collected in Microsoft Excel 2013 and analyzed in GraphPad Prism v7. All in vitro assays are expressed as mean ± standard deviation

(SD) and performed in triplicate. Data are compared using a two-sided unpaired $t$-test, one or two-way ANOVA with Tukey's, Sidak's, and Dunnett's multiple comparisons test. Tumor weights are expressed as mean ± SD and compared using Mann–Whitney tests; $p$-values < 0.05 were considered statistically significant. Rhythmicity was tested using the non-parametric test, JTK_Cycle, R version 4.0.3, package MetaCycle.

**Reporting summary**. Further information on research design is available in the Nature Research Reporting Summary linked to this article.

## Data availability
Publicly available clinical datasets GSE45547, GSE16476, and GSE85047. were analyzed with R2: Genomics Analysis and Visualization Platform (http://r2.amc.nl). RNA-

sequencing data generated in this study have been deposited in the European Nucleotide Archive (ENA), https://www.ebi.ac.uk/ena/), accession number: PRJEB36158. All Figures and Supplementary Figures have associated raw data. Raw data of Figs. 1–7 and Supplementary Fig. 1–18, and uncropped blots are provided as one Source Data file. All the other data are available within the article and its Supplementary Information. Source data are provided with this paper.

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

## Acknowledgements

We thank Brian J. Altman and Ronald Bernardi for proving SK-N-AS MYCN-ER and LAN5 ShMYCN cells, respectively. This project was supported by the Cell-Based Screening Service (C-BASS) core at BCM (core co-director: Dan Liu, Ph.D.), the Genomic and RNA Profiling Core at BCM (core directors: Dr. Lisa D. White, Ph.D., and Dr. Daniel Kraushaar, Ph.D.), and the Metabolomics Core at BCM (core director: Putluri Nagireddy, Ph.D.). E.B. is supported by the Alex's Lemonade Stand Foundation for Childhood Cancer, the St. Baldrick's Foundation, and the Cancer Prevention and Research Institute of Texas (CPRIT). G.P. is supported by Associazione Italiana per la Ricerca sul Cancro (Code: 15182). M.C. is supported by Fondazione Italiana per la Lotta al Neuroblastoma, Associazione Oncologia Pediatrica e Neuroblastoma, and Associazione Italiana per la Ricerca sul Cancro (Code: 19255). P.S. is partially funded by the European Union's Horizon 2020 research and innovation program under grant agreements 668858 and 826121. The Metabolomics Core at BCM is supported by CPRIT Core Facility Support Award RP170005 "Proteomic and Metabolomic Core Facility," NCI Cancer Center Support Grant P30CA125123, NIH/NCI R01CA220297, NIH/NCI R01CA216426, and intramural funds from the Dan L. Duncan Cancer Center (DLDCC).

## Author contributions

Conception and design: M.M.-S. and E.B. Development of methodology: L.T., B.Z, M.A. M., S.V., N.P., K.E.-M., G.P., and E.B. Acquisition of data: M.M.-S., G.M., L.T., M.A.M., S.D.G., R.B., K.R.K.R., B.F., J.H., and E.B. Analysis and interpretation of data: M.M.-S., L.T., M.A.M., M.C., P.S., N.P., T.P.B., and E.B. Writing, review, and/or revision of the manuscript: M.M.-S., L.T., K.E.-M., G.P., T.P.B., and E.B. Study supervision: E.B.

## Competing interests

The authors declare no competing interests.

## Additional information

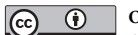

