## [Peer Review File · Nature Communications]

Reviewers' comments:

Reviewer #1 (Remarks to the Author):

It is known for a long time that the disruption of circadian rhythms in cancer correlates with the accelerated growth of tumor, and with the increase of the poor prognosis. Recently, evidences are accumulating that a proto-oncogene Myc-induced transcriptional cascade might directly disrupt the rhythm through E-box and non-E-box mediated pathways to downregulate Bmal1. This molecular machinery is very attractive for explaining how circadian signals contributes to tumorigenesis and tumor progression. In this study, Myrthala Moreno-Smith et al. further develop this hypothesis towards the clinical importance of correction of Myc-induced circadian and metabolic disturbances in neuroblastoma.

They first demonstrated that Rora and Bmal1 expression is strongly associated, and Rev-erb expression inversely correlated with clinical favorable outcome in three patient cohorts. Myc disrupts the clock by direct binding to Rev-erb and clock genes, and disrupts the circadian rhythms. Importantly, SR 1078, a selective agonist of ROR α and ROR γ , or over expression of Rora which stimulates ROR transcriptional activity, restores Bmal1 expression, and recovers the rhythm. Since ROR α stimulation inhibits the cell growth of neuroblastoma, they tried xenografts of this neuroblastoma cells under renal capsule of nude mouse, and found that SR1078 treatments impaired the tumor growth. Since RNAseq data outlined SR1078-mediated suppression of genes involved in lipid metabolism, and indeed the de novo lipogenesis is suppressed after the application of SR1078. Finally, authors demonstrated that SR1078 treatment markedly increased the anti-tumor activity of conventional chemotherapy.

This is a really interesting study of translational medicine utilizing the recent advances of molecular machinery of clock genes. I think that authors elegantly progress the Myc-circadian hypothesis on tumor oncology. Several few questions remained as follows.

Several questions:

1. Fig. 3D/E authors try to rescue the circadian expression of Bmal1 mRNA by Rora after serum shock on cells. However, in these analyses, the peak at 16-hours after serum-shock might just the reflection of the acute effect of serum-shock, without showing the true endogenous rhythms. So, it is better to perform a longer time course so that two full 24-hour cycles starting from 16h after serum shock are shown.

2. Fig 4D authors delineate alcohol and sterol pathway as most-suppressed pathway in GO analysis. However, authors did not further analyze this pathway, but instead authors then focus on three lipid biogenesis genes (Fig. 6). Were these genes among the differentially regulated genes in the RNAseq study (Fig. 4D)? If no, could the author more clearly justify why they are focusing on ACACA, FASN and SCD1 and not on genes that were among the "Sterol synthesis" pathway among down-regulated genes? .

Reviewer #2 (Remarks to the Author):

In the current study, the authors propose that the oncogene MYCN promotes the progression of neuroblastoma in part through inhibition of the circadian oscillator. MYCN appeared to suppress the expression BMAL1 and ROR α through the negative regulator REV-ERB α . Restoration of BMAL1 and circadian oscillation can inhibit neuroblastoma cell growth by repressing lipogenesis in those cells. Furthermore, activation of ROR α , a positive regulator of Bmal1, with a synthetic ligand blocks MYCN-mediated tumor growth and de novo lipogenesis and sensitizes NB tumors to chemotherapy.

The regulation of MYC on circadian clock and metabolism has been reported as the authors stated in introduction. The major contribution of this paper is the confirmation of a similar role for MYCN in neuroblastoma. While the repression of the positive regulatory arm of the molecular clock (including ROR α and BMAL1) by MYCN seems supported by evidence in cell culture, some concern arises over the limited scope of in vitro and in vivo experiments in supporting the proposed

mechanism.

Major comments

1. The authors used different cell lines in different figures without providing any rationale for why they changed the cell systems. While it's important to validate the key results in different cell lines, authors need to provide more details for the characteristics of each cell line and the reason to switch system for each experiment. In addition, MYCN protein levels need to be shown as this is the key player in the whole manuscript.
2. Many experiments lack proper controls. For example, in Fig. 2B, mRNA levels were compared during a time course of 4OHT treatment (to increase the activity of MYCN-ER) using time zero as a base. As circadian clock gene expression is known to change overtime (and can be synchronized by medium change), a 4OHT- control for each time point should be included. This applies to all figures, including tet-on/off and synthetic ligand treatments experiments.
3. Similarly, the RNA-seq and validation experiments in Fig. 4 need to be repeated with proper controls (not comparing 8 vs 24 hours). Otherwise, the results are meaningless.
4. Authors need to disclose how the ChIP-PCR primer sequences were determined. In other word, how the binding sites were identified. Mechanistically, how does MYCN suppress BMAL1/ROR α while enhancing REV-ERB α ? How does BMAL1 suppress lipogenic genes?
5. In figure 5, mice bearing non-MYCN amplified tumors (e.g. MYCN knockdown in the MYCN amplified cell lines or other NB cell lines where MYCN is not overexpressed) should be included as well to compare the effects of the ROR α agonist SR 1078 or ROR α transgenic activation on tumor burden. This will address to what extent the protective effect of ROR α activation is specific to MYCN activity in NB cells in vivo. In addition, the mice numbers for all tumor studies seem very small. The authors need to justify how they determine the statistical power for each cohort.

Minor point: BMAL1 is ARNTL, not ARNT!

Reviewer #3 (Remarks to the Author):

NATURE COMM REFEREE STATEMENT

General comments:

The study by Moreno-Smith et al. addresses the interesting subject of control of the cellular clock and its influence in cancer. Specifically, it explores the connection between MYCN-amplification and disruption of the clock in neuroblastoma, an area that has not been well studied so far. This work shows that MYCN can inhibit the cell clock, while the restoration of the clock by chemical or genetic intervention is able to reduce neuroblastoma cell growth and diminish lipogenesis. Specifically, they point out that MYCN strongly suppresses the clock central component BMAL1 and its activator ROR α , while it upregulates REV-ERB α (which inhibits BMAL1 transcription). Mechanistically, the authors show that MYCN binds BMAL1 and ROR α promoting regions blocking their transcription. Interestingly, the authors found that the ROR α / γ agonist SR1078 as well as overexpression of ROR α inhibit neuroblastoma cell survival, efficiently induce the expression of BMAL1, and the genes required for sterol synthesis. They also show that combined treatment of SR1078 and VP16 robustly inhibits tumor burden in a xenograft model.

The authors cite relevant previous publications in this subject and put their new findings into context. This is an innovative and important study which deepens our knowledge on the link between the molecular clock and MYCN in neuroblastoma. It also gives new insights on novel targets for developing therapies against MYCN-amplified neuroblastoma cases. Thus, this work is interesting for a broad readership and will merit publication in Nature Communications providing the following issues are addressed.

Major criticism:

- Please show growth curves of all the three in vivo experiments in Figure 5. In addition, please provide stainings of the tumors with some proliferation maker for instance Ki67, some apoptosis marker for instance Caspase 3, as well as MYCN, and if possible BMAL1. Indeed, the authors write in MoM that they prepare tumors for stainings however no stainings are shown. Please also provide weights of the animals throughout the experiment to give an indication that the treatments are not

toxic.

- Please include the NGP cells used in the in vivo experiments in MoM and provide growth curves for the xenograft experiment in Figure 7B. In addition, please provide stainings of the tumors with some proliferation marker for instance Ki67, some apoptosis marker for instance Caspase 3, as well as MYCN. Indeed, the authors write in MoM that they prepare tumors for stainings however no stainings are shown. Please also provide weights of the animals throughout the experiment to give an indication that the treatments are not toxic.
 - The authors check in several occasions cell viability using MTT or WST-8. These assays sense metabolic processes and are therefore not the best choice in a study that involves large metabolic changes. The authors need to compare an alternative method, like cell counting, with these assays, in order to conclude that they can be used in these conditions.
 - The authors claim to have used three NB datasets for patient data analysis. However, the Kocak and the SEQC contain the same patients. For the Kocak dataset the authors write n=498 while it is 649. They do not mention if they have removed any patients.
 - Please provide information on what happens with c-MYC/MYCN levels when restoring the clock.
-
- The authors claim that they performed a lipidomics analysis, which should be a large-scale study of pathways and networks of cellular lipids. Yet they solely reported triglycerides, diglycerides, C14:0 (myristic acid), C16:0 (palmitic acid), and C16:1 (palmitoleic acid). In addition, even though they asserted that MYCN-amplification increased the sterol synthesis, they did not analyze any sterol species (which is feasible with mass-based lipidomic approaches), while they just quantify the ratio between palmitoleic and palmitic acids. The authors therefore should implement the text with further descriptions coming from their lipidomic analysis
 - Importantly, it seems as the mass spectrometry data are missing, these should be included at least in the Supplemental material.
 - Similarly, the 13C labelled strategies are not described in the method section.
 - While the authors write that the analysis of C14:0 (myristic acid), C16:0 (palmitic acid), and C16:1 (palmitoleic acid) was performed by MS- spectrometry coupled with 13C labelled strategies, it is not clear how they quantify triglycerides and diglycerides. This information should be included in the main text.
 - Importantly, in the following experiments, only two biological replicates are shown: Figure 2A, 2B, 3A, 3B, 3C, 4D, 6B, and 7A. It is not possible to make statistical analysis with only two replicates. However, the authors declare to have performed t-test and gotten statistical significance. This is only possible because the authors consider the triplicate measurements of the each of the two biological replicates (which are just technical replicates and do not have biological value) as individual data points for their statistical analysis. This is incorrect. The authors need to average the triplicate values and use the average for each biological replicate for statistics. Thus, they need to perform and add additional biological replicates.
 - For the following experiments, the number of replicates is not indicated: Figure 3D, 3E, 5C, and 5G.
 - The authors have analyzed their in vivo data using the non-parametric tests Mann-Whitney and Kruskal-Wallis. Non-parametric tests are recommended only for data with non-Normal distribution. Have the authors performed a normality test on their data to conclude that it does not follow the Normal distribution? Generally, this type of data is Normal, and thus, the correct tests to be used are the parametric tests t-test and ANOVA. Non-parametric tests are less restrictive than parametric test, resulting in higher statistical significance values.
- Minor comments:
- A schematic summary of the elements of the molecular clock and their interactions would greatly help the reader to follow the text.
 - The authors should indicate the full composition of their vehicle for in vivo experiments. It is unlikely to use pure DMSO for injections in animals.
 - The authors should provide some information, based on bibliography or on their own experiments, regarding systemic toxicity of SR1078.
 - When referring to the gene, MYCN, italics is not used throughout the ms. Please revise.
-
- In Figure 1, MYCN is incorrectly written as "MYNC" in the graphs on the right; MYNC-non-amplified.
 - In the same figure, please harmonize the layout of the p-values, some are large, some small some in bold, some not.

- Please, revise the CypB bands in Figure 3B, LAN5-RORalpha cells. They do not seem to fit to the WB above since the bands are wider.
 - In several of the Figures, the asterisks shown have different sizes in the panels. Some are also almost hidden behind lines. See for instance Figure 3B and 6C.
 - In Figure 5B, does the CypB and BMAL1 data come from the same membrane? Due to the pattern of the bands this is unclear.
 - Figure 6G, could the authors provide more representative pictures of the cells? SR1078+oleic acid does not look as having a similar cell density as the control cells. Regarding this experiment, the control conditions with BSA alone is lacking.
 - Figure 6G. Please check "5x10⁵", the "x" does not seem to fit as it has a different style/font.
-
- From line 214 to line 218, the authors describe some negative results regarding p53 as "data not shown". Please, present this data as Supplemental Information. Additionally, the p53 mutation status of the cell lines used for those negative experiments might have an impact on the results. Please take this into account when interpreting the results.
 - On line 222 the authors write "alcohol biosynthetic processes". This expression should be revised as it is very vague and unclear
 - The authors should avoid overstatements like:
 - Line 226-227: "inhibits cell survival (...) by inhibiting the expression of genes required for sterol biosynthesis". Reduced sterol biosynthesis might be just a consequence of reduced cell growth, and not the cause.
 - Line 232-233: "restricts MYCN-mediated cell survival". Line 310-11: "Restoration of the clock effectively counteracts MYCN-mediated cell growth and metabolism". There is no evidence to support these affirmations. The restoration of the clock might have effects on additional pathways, leading to reduced cell proliferation, increased cell death and changes in metabolism (see previous comment) without directly interfering with MYCN signaling.
 - On line 272 there is a full stop missing.

Dear Editor and Reviewers,

We thank the Reviewers for expressing a high level of enthusiasm for our manuscript entitled "Restoration of the molecular clock is tumor suppressive in neuroblastoma" (NCOMMS-17-23200A-Z), and providing insightful comments and suggestions. We have addressed all of the reviewers' concerns and suggested studies as summarized point by point below and highlighted in gray throughout the text.

Reviewer #1:

Critique 1: Fig. 3D/E authors try to rescue the circadian expression of *Bmal1* mRNA by *Rora* after serum shock on cells. However, in these analyses, the peak at 16-hours after serum-shock might just be the reflection of the acute effect of serum-shock, without showing the true endogenous rhythms. So, it is better to perform a longer time course so that two full 24-hour cycles starting from 16h after serum shock are shown.

Response: As suggested, we performed a longer time course experiment, analyzing *BMAL1* mRNA expression every six hours for two full 24h cycles starting from 16h after serum shock. The rescue of endogenous *BMAL1* oscillation is now shown in the **new Figure 3D**. To confirm rhythmicity in gene expression, we performed the non-parametric JTK rhythmicity test, which revealed a large increase in circadian amplitude (new Figure 3D, right table).

Critique 2: Fig 4D authors delineate alcohol and sterol pathway as most-suppressed pathway in GO analysis. However, authors did not further analyze this pathway, but instead authors then focus on three lipid biogenesis genes (Fig. 6). Were these genes among the differentially regulated genes in the RNAseq study (Fig. 4D)? If no, could the author more clearly justify why they are focusing on *ACACA*, *FASN* and *SCD1* and not on genes that were among the "Sterol synthesis" pathway among down-regulated genes?

Response: Our RNA-seq data in NB cells (**new Figure 4D**) showed that two of the most downregulated biological processes were cholesterol biosynthesis and regulation by SREBP (adjusted $p=2.44E^{-09}$), and metabolism of lipids and lipoproteins (adjusted $p=1.01E^{-03}$). We further selected 10 genes significantly altered by SR1078 and specifically involved in both cholesterol synthesis (*IDI-1*, *HMGCR*, *HMGCS1*, and *MVK2*) and lipid biogenesis (*FABP3*, *FABP5*, *ELOVL2*, *ELOVL6*, *ACSL3*, and *SCD1*; *ACACA* and *FASN* did not reach significance) and validated their expression upon SR1078 by q-PCR in MNA cells (**new Supplementary Figure 11**). We did not further investigate the cholesterol synthesis pathway because the cholesterol esters were increased in the xenograft tumors treated with SR1078 (**new Supplementary Figure 14**). In contrast, the intratumoral levels of glycerolipids, such as triglycerides (TGs) and their precursors diglycerides (DGs) were significantly reduced by SR1078, suggesting that activation of *ROR α* effectively inhibits *in vivo* lipogenesis (Figure 5C and **new Supplementary Figure 14**). To further address this point and because fatty acids (FAs) are required for DG and TG synthesis, we decided to characterize the FA composition induced by SR1078 in the xenograft tumors by LC-MS (**new Figure 5D**). Consistently with our RNA-seq (**new Figure 4D**) and lipidomics data (Figure 5C), we found that SR1078 significantly reduces the levels of intratumoral FAs including C12:0, C16:0, C18:1, and C20:3, which serve as side chains of the reduced DG and TG groups (**new Figure 5D and new Supplementary Table 4**), altogether suggesting that SR1078 blocks tumor growth by inhibiting FA and glycerolipid biosynthesis. Because *ACACA* and *FASN* are key enzymes in FA synthesis and *SCD1* functions in FA desaturation, we then focused on evaluating MYCN and *BMAL1* regulation of lipogenic gene expression, and *de novo* lipogenesis and desaturation activity (**new Figures 6C and 6E**). These new data are now included in the Results section and discussed in the Discussion section.

Reviewer #2:

Critique 1: The authors used different cell lines in different figures without providing any rationale for why they changed the cell systems. While it's important to validate the key results in different cell lines, authors need to provide more details for the characteristics of each cell line and the reason to switch

system for each experiment. In addition, MYCN protein levels need to be shown as this is the key player in the whole manuscript.

Response: To elucidate the regulation of MYCN on clock genes expression, we decided to use different inducible cell systems in which MYCN expression/activation could be selectively modulated. Two systems were employed for MYCN activation: MYCN3 (MYCN Tet-ON) and SK-N-AS MYCN-ER; and two for MYCN inhibition: LAN5 ShMYCN and TET-21/N (MYCN Tet-OFF). The details and the rationale for the selected lines are now included in the Results section. In addition, MYCN protein levels are now shown in the **new Figure 2A and Supplementary Figure 7**.

Critique 2: Many experiments lack proper controls. For example, in Fig. 2B, mRNA levels were compared during a time course of 4OHT treatment (to increase the activity of MYCN-ER) using time zero as a base. As circadian clock gene expression is known to change overtime (and can be synchronized by medium change), a 4OHT- control for each time point should be included. This applies to all figures, including tet-on/off and synthetic ligand treatments experiments.

Response: We agree with the Reviewer that circadian gene expression changes overtime. As suggested, we have now repeated our gene expression study including a 4OHT- control for each time point (**new Figure 2B**). Regarding synthetic ligand treatment, gene expression was determined at one specific time point (Figures 3A and 3C). In addition, we have now demonstrated that both doxycycline (DOX) and 4OHT do not alter clock gene expression (*DBP*, *BMAL1* and *PER2*) in parental LAN5 and SK-N-AS cells, as shown in the **new Supplementary Figure 2**.

Critique 3: Similarly, the RNA-seq and validation experiments in Fig. 4 need to be repeated with proper controls (not comparing 8 vs 24 hours). Otherwise, the results are meaningless.

Response: As suggested, the RNA-seq and validation studies were repeated and re-analyzed comparing SR1078 8h exposure to its corresponding CTRL 8h. The new heap map and pathway enrichment analysis are now included in the **new Figure 4D**. Similarly, gene expression validation comparing SR1078 8h to CTRL 8h is now included in the **new Supplementary Figure 11B**.

Critique 4: Authors need to disclose how the ChIP-PCR primer sequences were determined. In other word, how the binding sites were identified. Mechanistically, how does MYCN suppress BMAL1/ROR α while enhancing REV-ERB α ? How does BMAL1 suppress lipogenic genes?

Response: The graphical representation of the genomic regions analyzed in ChIP-qPCR assays is shown in the Supplementary Figure 4. MYCN binding sites were identified as follows: canonical MYCN binding site on *REV-ERB α* gene promoter (-500bp from TSS-NM_021724.5/Hg38) was identified by *in silico* analysis; several PCR amplicons in the *ROR α* 1/4 TSS surrounding regions were tested (see Supplementary Methods) and positive amplicons in the TSS region of both gene promoters were identified; BMAL1 ChIP-qPCR primers were described by Shostak, et al. (PMID: 27339797; reference 19). MYCN induces transcriptional repression by an indirect binding to DNA in part via MIZ1. To address the question on how MYCN suppresses the positive arm of the molecular clock (main finding), we performed MYCN ChIP-qPCR analysis in MNA cells upon genetic depletion of MIZ1 (LAN5 ShMIZ1). Effective knockdown of MIZ1 (**new Supplementary Figures 5A and B**) significantly reduces MYCN binding at the promoter regions of *BMAL1* and *ROR α* (**new Supplementary Figure 5C and D**), suggesting that MYCN repression of the clock requires MIZ1. The molecular mechanisms through which MYCN directly activates REV-ERB α , the negative regulator of BMAL1, have already been elucidated in detail by Altman, et al. (2015, *Cell Metabolism*; 22, 1–11).

Because depletion of BMAL1 completely rescues the inhibition of lipogenic gene expression mediated by ROR α activation (Figure 6B), we looked at BMAL1 binding at the promoter regions of lipogenic genes upon MYCN activation, ROR α overexpression, and these conditions combined to determine whether changes in expression were mediated by changes in the binding of BMAL1 directly to these loci. ChIP-qPCR analysis in SK-N-AS MYCN-ER ROR α cells confirmed reduced BMAL1 binding at promoters of all three lipogenic enzymes upon activation of MYCN and increased binding upon overexpression of ROR α

(**new Figure 6C**). Moreover, overexpression of ROR α completely rescues BMAL1 binding, suggesting that MYCN and BMAL1 compete for binding at these loci. These new data are now included in the Results section and discussed in the Discussion section.

Critique 5: In figure 5, mice bearing non-MYCN amplified tumors (e.g. MYCN knockdown in the MYCN amplified cell lines or other NB cell lines where MYCN is not overexpressed) should be included as well to compare the effects of the ROR α agonist SR1078 or ROR α transgenic activation on tumor burden. This will address to what extent the protective effect of ROR α activation is specific to MYCN activity in NB cells *in vivo*. In addition, the mice numbers for all tumor studies seem very small. The authors need to justify how they determine the statistical power for each cohort.

Response: To address whether the anti-tumor effect of ROR α activation is specific to MYCN activity, we compared the effect of SR1078 on tumor growth in MYCN-amplified vs. non-amplified orthotopic xenografts. SR1078 failed to block tumor growth in non-MYCN amplified SK-N-AS-derived xenografts (**new Figure 5B**), suggesting that the protective effect of ROR α activation via SR1078 is indeed MYCN-specific. The mice numbers for this new study were CTRL group n=10 and SR1078 group n=10. We agree with the Reviewer that the activity of SR1078 was previously tested in a limited number of IMR32-derived xenografts. As suggested, we repeated this study including more mice in each group (CTRL group n=11 and SR1078 group n=11). Our new data (**new Figure 5A**) show the dramatic effect of SR1078 in controlling MYCN-driven tumor growth. The statistical power is now robustly enhanced ($p=0.00038$, Mann-Whitney test) and all the statistical calculations are now included both in the Results section and Figure Legends.

Minor point: BMAL1 is ARNTL, not ARNT

Response: Thanks, this was corrected.

Reviewer #3:

Critique 1: Please show growth curves of all the three *in vivo* experiments in Figure 5. In addition, please provide stainings of the tumors with some proliferation maker for instance Ki67, some apoptosis marker for instance Caspase 3, as well as MYCN, and if possible BMAL1. Indeed, the authors write in M&M that they prepare tumors for stainings however no stainings are shown. Please also provide weights of the animals throughout the experiment to give an indication that the treatments are not toxic.

Response: We did not obtain growth curves for the *in vivo* experiments of Figure 5. Luc flux data don't usually accurately reflect tumor growth in our orthotopic model due to the extended necrotic areas present in the tumors at a later stage. The endpoint assessment of tumor weights is a very reliable and direct method to evaluate the effect on tumor growth in our model. As suggested, Ki67 and caspase 3 staining were performed in IMR32 and LAN5-ROR α tumor samples at the end of treatment. Percentages of Ki67 and caspase positive cells from tumors both treated with SR1078 and overexpressing ROR α are presented in the **new Supplementary Figure 13**. The body weights and the health general conditions of the mice were recorded weekly throughout the studies and no apparent signs of toxicity related to SR1078 treatment were noted. As suggested, the body weights from controls and SR1078 xenografts derived from both MYCN-amplified and non-MYCN amplified lines are now compared in the **new Supplementary Figure 12**.

Critique 2: Please include the NGP cells used in the *in vivo* experiments in MoM and provide growth curves for the xenograft experiment in Figure 7B.

Response: Thanks, NGP cells are now included in the Methods section. No tumor growth curves were obtained from Figure 7B (see response to critique 1).

Critique 3: The authors check in several occasion cell viability using MTT or WST-8. These assays sense metabolic processes and are therefore not the best choice in a study that involves large metabolic

changes. The authors need to compare an alternative method, like cell counting, with these assays, in order to conclude that they can be used in these conditions.

Response: As suggested, we compared MTT and CCK-8 cell viability assays to cell counting in two MYCN-amplified cell lines (LAN5 and IMR32) and in LAN5 ROR α cells. Data from MTT and CCK-8 assays nicely correlate with those generated by cell counting, and confirmed the same changes in cell proliferation (**new Supplementary Figure 9** and updated Results section).

Critique 4: The authors claim to have used three NB datasets for patient data analysis. However, the Kocak and the SEQC contain the same patients. For the Kocak dataset the authors write n=498 while it is 649. They do not mention if they have removed any patients.

Response: We thank the Reviewer for raising this point. The Kocak dataset includes 649 samples annotated with INSS stage, age at diagnosis, and MYCN status; however, only 476 are annotated with survival data. We have now clarified this point in the Methods section. Because the Kocak and SEQC cohorts partially overlap, we have now replaced the SEQC dataset with the most recent published NRC dataset (GSE85047) composed of 283 samples (**new Supplementary Figure 1C** and updated Results section).

Critique 5: Please provide information on what happens with c-MYC/MYCN levels when restoring the clock.

Response: To determine whether restoration of the clock alters c-MYC and MYCN expression, we looked at c-MYC and MYCN mRNA and protein levels upon SR1078 treatment. MYCN protein stabilization was notably decreased by SR1078 in MNA cells (LAN5 and SK-N-BE(2)C); however, no changes in c-MYC protein levels were detected in non-MNA c-MYC overexpressing cells (SH-SY5Y) (**new Supplementary Figure 7** and updated Results section), suggesting that this regulation is MYCN-specific.

Critique 6: The authors claim that they performed a lipidomics analysis, which should be a large-scale study of pathways and networks of cellular lipids. Yet they solely reported triglycerides, diglycerides, C14:0 (myristic acid), C16:0 (palmitic acid), and C16:1 (palmitoleic acid). In addition, even though they asserted that MYCN-amplification increased the sterol synthesis, they did not analyze any sterol species (which is feasible with mass-based lipidomic approaches), while they just quantify the ratio between palmitoleic and palmitic acids. The authors therefore should implement the text with further descriptions coming from their lipidomic analysis.

Response: Our revised manuscript includes now the complete *in vivo* lipidomics analysis, which contains the cholesterol esters (**new Supplementary Figure 14** and updated Results and Discussion sections). This point has been mostly discussed in the response to critique 2 of Reviewer #1.

Critique 7: The ¹³C labelled strategies are not described in the method section.

Response: The FA tracing strategies, including deuterated water and ¹³C labelling, are described in the Method section under FA measurement.

Critique 8: While the authors write that the analysis of C14:0 (myristic acid), C16:0 (palmitic acid), and C16:1 (palmitoleic acid) was performed by MS- spectrometry coupled with ¹³C labelled strategies, it is not clear how they quantify triglycerides and diglycerides. This information should be included in the main text.

Response: Lipidomics processing and statistical analysis have been largely described by Vantaku et al., Clin. Cancer Res., 2019. We have now included DG and TG quantification methods and relevant references in the Methods section under Lipidomics.

Critique 9: Importantly, in the following experiments, only two biological replicates are shown: Figure 2A, 2B, 3A, 3B, 3C, 4D, 6B, and 7A. It is not possible to make statistical analysis with only two replicates.

However, the authors declare to have performed t-test and gotten statistical significance. This is only possible because the authors consider the triplicate measurements of the each of the two biological replicates (which are just technical replicates and do not have biological value) as individual data points for their statistical analysis. This is incorrect. The authors need to average the triplicate values and use the average for each biological replicate for statistics. Thus, they need to perform and add additional biological replicates.

Response: As requested, we performed a third biological experiment for Figures 2A, 2B, 3A, 3C and 6B. These new data are now presented in the **new Figures 2A, 2B, 3A, 3C and 6B**. For data analysis, we first averaged triplicate values of each biological experiment. Then, we used the averages from the three biological experiments for statistics. Figure 3B shows averages data from two biological WB experiments and statistical analysis is not included. q-PCR validation data of Figure 4D have been moved to the new Supplementary Figure 11B and are mean \pm SD from three biological replicates. Figure 7A includes data from one of two biological experiments (n=3) and statistical analysis is not included. The number of biological replicates has now been updated in the new Figure Legend section.

Critique 10: The authors have analyzed their *in vivo* data using the non-parametric tests Mann-Whitney and Kruskal-Wallis. Non-parametric tests are recommended only for data with non-Normal distribution. Have the authors performed a normality test on their data to conclude that it does not follow the Normal distribution? Generally, this type of data is Normal, and thus, the correct tests to be used are the parametric tests t-test and ANOVA. Non-parametric tests are less restrictive than parametric test, resulting in higher statistical significance values.

Response: Due to the variability of single mouse data, all the *in vivo* studies were analyzed using the non-parametric test Mann-Whitney, which makes fewer assumptions about data distribution. We believe the correct method was applied to our analyses.

Critique 11: A schematic summary of the elements of the molecular clock and their interactions would greatly help the reader to follow the text.

Response: Thanks for the suggestion. This is now included in the **new Supplementary Figure 17**.

Critique 12: The authors should indicate the full composition of their vehicle for *in vivo* experiments. It is unlikely to use pure DMSO for injections in animals.

Response: The vehicle composition is now included in the updated Methods section.

Critique 13: The authors should provide some information, based on bibliography or on their own experiments, regarding systemic toxicity of SR1078.

Response: As discussed in critique 1, this information is now included in the **new Supplementary Figure 12**.

Critique 14: When referring to the gene, MYCN, italics is not used throughout the ms. Please revise.

Response: The gene nomenclature is now corrected.

Critique 15: In Figure 1, MYCN is incorrectly written as "MYNC" in the graphs on the right; MYNC-non-amplified.

Response: Thanks, we have now corrected and reformatted Figure 1 (**new Figure 1**).

Critique 16: Please, revise the CypB bands in Figure 3B, LAN5-RORalpha cells. They do not seem to fit to the WB above since the bands are wider.

Response: We thank the Reviewer for noticing this error. This has now been corrected in the **new Figure 3B**. The full blots can be found in the Source Data file.

Critique 17: In several Figures, the asterisks have different sizes in the panels. Some are also almost hidden behind lines. See for instance Figure 3B and 6C.

Response: All asterisks have now been corrected.

Critique 18: In Figure 5B, does the CypB and BMAL1 data come from the same membrane? Due to the pattern of the bands this is unclear.

Response: Figure 5B is now Figure 5C. We carefully reviewed the full blot from Figure 5C. We confirm that CypB and BMAL1 come from the same membrane. The full blots can be found in the Source Data file.

Critique 19: Figure 6G could the authors provide more representative pictures of the cells? SR1078+oleic acid does not look as having a similar cell density as the control cells. Regarding this experiment, the control conditions with BSA alone is lacking.

Response: Three biological experiments were performed including control with BSA. New representative images from all the three conditions (CTRL BSA, SR1078, and SR1078+oleic acid) are now included in the **new Figure 6H**.

Critique 20: Figure 6G. Please check "5x10⁵", the "x" does not seem to fit as it has a different style/font.

Response: Font/style of "x" is now corrected.

Critique 21: From line 214 to line 218, the authors describe some negative results regarding p53 as "data not shown". Please, present this data as Supplemental Information. Additionally, the p53 mutation status of the cell lines used for those negative experiments might have an impact on the results. Please take this into account when interpreting the results.

Response: We tested the effect of SR1078 on p53 protein expression in three MYCN-amplified p53 wild-type cell lines: LAN5, IMR32, and Kelly. The effect of SR1078 on cell growth was also assessed in LAN5 cells upon genetic depletion of p53 (LAN5 Shp53). All these new data are now included in the updated Results section and **new Supplementary Figure 10**.

Critique 22: On line 222 the authors write "alcohol biosynthetic processes". This expression should be revised as it is very vague and unclear

Response: This statement is no longer present. The RNA-seq data were re-acquired and re-analyzed (**new Figure 4D**).

Critique 23: The authors should avoid overstatements like: Line 226-227: "inhibits cell survival (...) by inhibiting the expression of genes required for sterol biosynthesis". Reduced sterol biosynthesis might be just a consequence of reduced cell growth, and not the cause.

Response: This statement is no longer present. The RNA-seq data were re-acquired and re-analyzed (**new Figure 4D**).

Critique 24: Line 232-233: "restricts MYCN-mediated cell survival". Line 310-11: "Restoration of the clock effectively counteracts MYCN-mediated cell growth and metabolism". There is no evidence to support these affirmations. The restoration of the clock might have effects on additional pathways, leading to reduced cell proliferation, increased cell death and changes in metabolism (see previous comment) without directly interfering with MYCN signaling.

Response: We agree with the Reviewer that restoration of the clock could potentially interfere with additional cellular functions and processes. However, collectively, our manuscript by employing *in vitro* and *in vivo* NB MYCN models show that BMAL1 activation competes with MYCN-driven oncogenesis and cell metabolism. Moreover, our finding that MYCN interferes with BMAL1 target activation strongly

suggests that one of the central mechanisms by which BMAL1 inhibits cell growth is by inhibiting MYCN function (new Figure 6C and Supplementary Figure 17). We have added an additional statement, suggesting that while we believe this to be a primary mechanism by which MYCN-mediated cell growth and metabolism is inhibited, there may be some contribution by other clock-regulated pathways.

Critique 25: On line 272 there is a full stop missing.

Response: This is now corrected.

REVIEWERS' COMMENTS

Reviewer #1 (Remarks to the Author):

All the questions are adequately answered, and now the manuscript became much improved. By initiating transcriptome of patient cohorts, authors elegantly expand the MYCN-circadian hypothesis from circadian clock machinery to neuroblastoma treatment. It clearly demonstrates that restoration of clock really suppresses the tumor. This is a nice translational work in high-risk neuroblastoma, and I want to hear the outcome of further clinical trial by utilizing this data.

Reviewer #2 (Remarks to the Author):

The authors have addressed my main concerns.

Reviewer #3 (Remarks to the Author):

The authors have addressed all points from the referees and substantially improved their manuscript.